# AssoMem: Scalable Memory QA with Multi-Signal Associative Retrieval

**Kai Zhang**[1, 2, †]**, Xinyuan Zhang**[2, †]**, Ejaz Ahmed**[2]**, Hongda Jiang**[2]**, Caleb Kumar**[2]
**Kai Sun**[2]**, Zhaojiang Lin**[2]**, Sanat Sharma**[2]**, Shereen Oraby**[2]**, Aaron Colak**[2]
**Ahmed Aly**[2]**, Anuj Kumar**[2]**, Xiaozhong Liu**[1]**, Xin Luna Dong**[2, †]

[1]Worcester Polytechnic Institute, [2]Meta Reality Labs
[†]{kkaizh, dylanz426, lunadong}@meta.com

## Abstract

Accurate recall from large-scale memories remains a core challenge for memory-augmented AI assistants performing question answering (QA), especially in similarity-dense scenarios where existing methods mainly rely on semantic distance to the query for retrieval. Inspired by how humans link information associatively, we propose **AssoMem**, a novel framework constructing an **asso**ciative **mem**ory graph that anchors dialogue utterances to automatically extracted *clues*. This structure provides a rich organizational view of the conversational context and facilitates importance-aware ranking. Further, AssoMem integrates multi-dimensional retrieval signals—*relevance*, *importance*, and *temporal* alignment—using an adaptive mutual information (MI)-driven fusion strategy. Extensive experiments across three benchmarks and a newly introduced dataset, MeetingQA, demonstrate that AssoMem consistently outperforms state-of-the-art baselines, verifying its superiority in context-aware memory recall.

## 1 Introduction

The rapid advancement of large language models (LLMs) has opened the door to personal assistants that function as a "second brain"—a digital companion capable of capturing, organizing, and retrieving information on behalf of the user. An essential capability of such systems is the ability, with explicit user consent, to store and recall events and facts from the user's life (Jiang et al., 2025). This capability enables natural memory-recall interactions such as "Summarize the key points from my meeting with Sarah yesterday". In this paper, we focus on settings where *textual* memories such as meeting notes, and conversational dialogues are continuously accumulated over time, and we study the problem of *answering memory recall questions over large-scale memory repositories*.

Recent research in this area predominantly follows the Retrieval-Augmented Generation (RAG) paradigm, organizing user memories to enable efficient retrieval and accurate response generation.

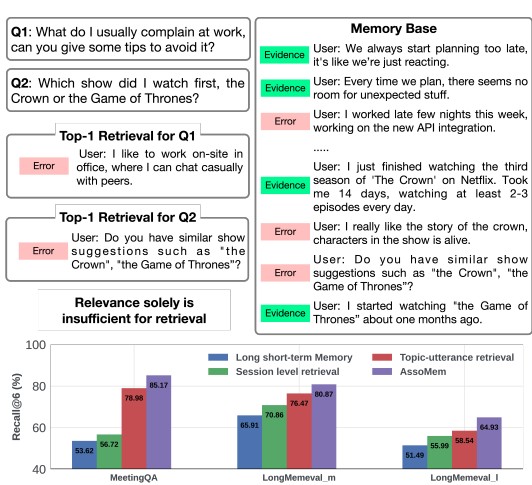

Figure 1: An example showing limitations in relevance solely retrieval. Our AssoMem consistently outperforms SOTA baselines on three datasets.

Drawing inspiration from the effectiveness of human memory systems in recalling information from *sequential streams*, several approaches partition historical dialogues into long- and short-term segments to enhance memory recall (Zhang et al., 2024; Li et al., 2025; Zhong et al., 2024; Chhikara et al., 2025). Other methods organize memory using hierarchical filters—such as topics or summaries—to narrow down the retrieval search space (Tan et al., 2025; Xu et al., 2025b;

Zhou et al., 2025). Graph-based approaches construct an entity-relationship knowledge graph from personal memories and answer questions through graph algorithms (Wang et al., 2024; Chhikara et al., 2025). Despite these advances in memory structure optimization, a critical challenge remains: as the volume of user memory records increases, retrieval performance deteriorates as in Figure 1, largely because the memory pool accumulates many highly similar items, like repeated meeting topics and overlapping conversation snippets, making it harder to distinguish truly relevant information.

We argue that humans do not perceive their memories as isolated entries or as a simple chronological stream. Instead, they organize them *associatively*, linking pieces of information through *clues* such as entities, locations, events, and topics. Likewise, pinpointing relevant memory evidence cannot rely solely on refining relevance comparison, as explored in reranking modules (Tan et al., 2025) or multi-granularity retrieval (Xu et al., 2025a). Among related items, people tend to remember *important clues* more clearly and revisit them more often; for example, answering "What do I usually complain at work, can you give some tips?" requires identifying the clues that matter most to the user as depicted in Figure 1.

Motivated by these observations, we propose ***AssoMem***, a memory QA framework that leverages associative structures to guide memory selection. At its core is an *associative memory graph* that links each memory utterance to a set of *clues*, which are automatically extracted by LLMs to enable fine-grained interpretation of a user's memories and to connect memories sharing similar signals. Unlike existing memory graphs that are built entirely on abstractive concepts rather than raw historical data (Wang et al., 2024; Chhikara et al., 2025; Xu et al., 2025b), this graph supports associative connections from abstractive clues to exact memories, facilitating *importance-aware ranking*. Based on this graph, AssoMem further integrates multiple retrieval signals—*relevance*, *importance*, and *temporal* alignment—and employs a mutual-information (MI)–driven fusion strategy to dynamically balance these dimensions according to query intent, yielding more accurate and context-aware memory retrieval. Moreover, fine-tuning the answer generation model with a multi-task denoising strategy is utilized to maximize the QA performance based on retrieved memory records. To the best of our knowledge, AssoMem is the first memory QA system that mimics the structure of associative memory to enhance QA on large-scale, similarity-dense memory collections. We summarize our main contributions as follows:

- **Unified framework for memory recall QA:** We propose AssoMem, a memory QA system that integrates *relevance*, *importance*, and *temporal* signals through a mutual information (MI)-driven weight assignment strategy, enabling adaptive and context-aware memory selection to improve answer quality.
- **Associative memory graph**: At the core of AssoMem is an associative memory graph that captures semantic relationships between utterances and the clues associated with them. This graph facilitates efficient retrieval and importance-aware ranking of memories.
- **New benchmark and evaluation:** To foster research in large-scale memory retrieval, we introduce **MeetingQA**, a benchmark simulating real-world meeting scenarios where multi-turn dialogues form the memory base, paired with diverse QA examples. Extensive experiments and ablations on MeetingQA and other memory benchmarks show that AssoMem outperforms existing retrieval approaches by **24.93%** on average.

## 2 RELATED WORK

**Large-scale Memory Management and Retrieval** Memory has emerged as a promising solution for enhancing LLMs (Madaan et al., 2022; Wang et al., 2023). However, as the historical memory incrementally accumulates, existing methods fail to process the large-scale memory since the incremental information poses noise for both retrieval and generation(Yu et al., 2025; Hu et al., 2025; Maharana et al., 2024). Recent research in this area has advanced along two key aspects: large-scale memory management and retrieval. In terms of large-scale memory management, early work introduced structural organization by partitioning conversational history into short- and long-term memory (Zhang et al., 2024; Li et al., 2025; Zhong et al., 2024), enabling models to reflect both recent interactions and persistent user preferences. More recent approaches incorporate hierarchical structures—such as topics, summaries, and memory graphs—to constrain the retrieval space and improve recall performance (Tan et al., 2025; Xu et al., 2025b; Chhikara et al., 2025; Rezazadeh et al., 2024). In terms of retrieval, focus is on directly identifying relevant memories for downstream tasks. *Query-centered* methods enhance retrieval through improved query formulation

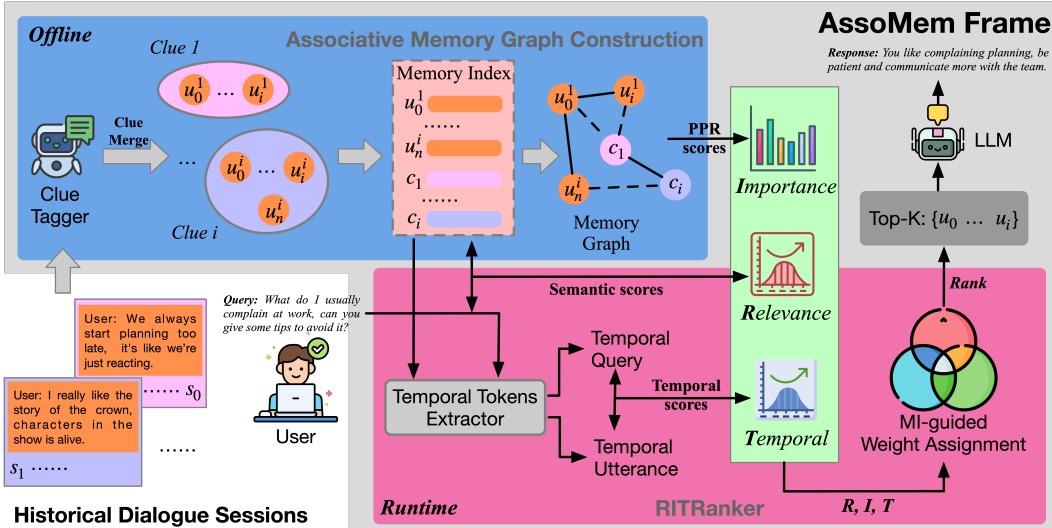

Figure 2: Overview of the proposed AssoMem framework. A topic–utterance graph is constructed from historical dialogues, enabling the integration of relevance, importance, and temporal signals. These are adaptively fused to guide accurate memory retrieval for question answering.

(Jiang et al., 2023; Jang et al., 2024), while reranking-based approaches refine retrieved candidates using trainable scoring mechanisms (Wu et al., 2024b; Du et al., 2024; Tan et al., 2025). Complementary works have also explored granularity's impacts, demonstrating the effectiveness of hybrid retrieval strategies (Xu et al., 2025a; Tan et al., 2025; Sarthi et al., 2024). Despite their success, these methodologies are predominantly grounded in *relevance*, aiming to retrieve the most topically similar memories while overlooking key challenges discussed in Section 1. In contrast, AssoMem adopts an associative memory to integrate multi-dimensional signals to address these challenges.

**PageRank** (PR) models a random surfer over a directed graph, where node scores reflect the stationary distribution of the walk, yielding an *importance* prior that complements term-matching relevance in web search (Page et al., 1999; Brin & Page, 1998; Langville & Meyer, 2011). Alongside HITS (Kleinberg, 1999), it underpinned early large-scale retrieval by propagating hyperlink endorsements and proved effective under sparse or noisy lexical signals. Subsequent works improved computational scalability via power-method accelerations and linear-algebraic solvers (Bahmani et al., 2010). Personalized PageRank (PPR) biases teleportation toward user-specific needs, effectively yielding a relevance-conditioned importance signal (Bahmani et al., 2010; Wayama & Sugiyama, 2025). In our work, AssoMem leverages PPR to enable multi-dimensional signals for later retrieval.

## 3 METHODOLOGY

### 3.1 PROBLEM FORMULATION AND SOLUTION FRAMEWORK

**Problem formulation** Consider a *memory bank* $\mathcal{M} = \{(S_0, d_0), (S_1, d_1), \ldots, (S_N, d_N)\}$, where each *session* $S_i = \{u_0, u_1, \ldots, u_n\}$ contains $n$ *utterances* and is associated with a timestamp $d_i$. A question $q$ is considered a *memory question* if it specifically refers to the user's past, as illustrated in Figure 1. The *memory recall* problem takes a memory question $q$ and outputs an answer based on the memory bank $\mathcal{M}$. Note that a memory question can go beyond specific memory seeking such as "can you give some tips...". While our method can apply to *personalized conversations* where the user does *not* explicitly refer to the past, we primarily focus on memory recall questions in this paper.

**Solution framework** At run time, we answer memory questions in two steps as depicted in Figure 2: memory retrieval and answer generation. Given a question $q$, the *Retrieval* step retrieves a set of memory utterances to ground question answering, denoted by $\mathcal{E}^*$; the *Answer Generation* step then generates the answer to $q$ based on the retrieved memories: $\hat{a} = \text{LLM}^*(q, \mathcal{E}^*)$.

## 3.2 MEMORY RETRIEVAL

The retrieval step aims to select the best memory evidence to support QA. At the core of our retrieval is the *Associative Memory Graph* that anchors each piece of memory with the underlying clues, and connects relevant memories. We next discuss the construction of the graph and how we use it for the downstream QA task.

### 3.2.1 ASSOCIATIVE MEMORY GRAPH CONSTRUCTION

**Memory clues** A memory clue captures potential cues that help memorization; it can be an aspect phrase like *Evening entertainment*, a key entity like *Lumia project*, or an event like *launching weekly meetings*. For each session $S_i$ in the memory bank, we employ an LLM agent to generate a representative clue $c_i$ for the session. This clue is associated with all utterances in $S_i$, resulting in an initial clue set $\mathcal{C} = \{c_0, c_1, \ldots, c_N\}$, where $N$ is the number of sessions.

To reduce redundancy and enhance topic coherence, we merge clues with high semantic similarity. Specifically, for any pair of clues $(c_i, c_j)$, if their embedding similarity exceeds a threshold $\delta$, they are merged into the same clue. The associated utterances from the merged clues are grouped under the new clue. This process yields a refined clue set $\mathcal{C}'$ and updated utterance groupings.

**Associative Memory Graph** An associative memory graph $\mathcal{G} = (\mathcal{V}, \mathcal{E})$ associates each memory record with the underlying clues and other relevant memories. There are two types of nodes: each *clue node* represents a clue in the set $\mathcal{C}'$ of merged clues, and each *utterance node* represents an utterance in $\mathcal{U}$. Each node (clue or utterance) is represented by its text embedding, computed using a pre-trained embedding model (e.g., BGE). There are also two types of edges: each *ownership edge* connects an utterance $u \in S_i$ with its associated clue $c_i'$, and each *similarity edge* connects clues or utterances that are similar. Specifically, we create an edge for a pair of clues or a pair of utterances whose embedding similarity exceeds a pre-decided threshold $\gamma$:

$$\text{sim}(v_i, v_j) > \gamma, \quad v_i, v_j \in \mathcal{C}' \text{ or } v_i, v_j \in \mathcal{U} \tag{1}$$

The structure can be extended with conversational metadata (e.g., people, locations) when it exists and enriched via external taxonomies such as ConceptNet (Speer et al., 2017).

### 3.2.2 CANDIDATE RETRIEVAL

Inspired by how associative memory works, we perform candidate retrieval in a two-step hybrid mode. First, given a query $q$, we retrieve the relevant clues, and denote the Top-$K$ clues related to $q$ by $\mathcal{C}_q$; we consider all utterances associated with $\mathcal{C}_q$ as candidate memories, denoted by $\mathcal{U}_{\text{cand}} = \{u \mid \text{clue}(u) \in \mathcal{C}_q\}$. Next, we rank all candidate utterances by a score predicting their usefulness in answering the question, obtaining the final retrieval results: $\mathcal{E}^* = \underset{\mathcal{E} \subseteq \mathcal{U}_{\text{cand}}, |\mathcal{E}| = K}{\arg\max} \sum_{u \in \mathcal{E}} \text{Score}(q, u)$.

With a good scoring system, denoted by $\text{Score}(q, u)$, this hybrid retrieval strategy returns accurate memory evidence and ensures that $\mathcal{E}^*$ maximizes the utility of the retrieved memories. Designing the scoring system plays a critical role to QA quality, which we will describe in detail next.

### 3.2.3 RITRANKER: RELEVANCE, IMPORTANCE, AND TEMPORAL DYNAMICS

Unlike existing memory-based methods that rely solely on similarity, our retrieval score for each utterance $u$ integrates three dimensions: *relevance*, *importance*, and *temporal alignment*. This fusion enables retrieval of memories that not only align with the recall question but also reflect central aspects of the user's daily life and adhere to temporal constraints.

**Relevance** Existing memory-based methods have verified that semantic relevance serves as an essential criteria and ensures that retrieved utterances are contextually aligned with the query. We compute relevance using cosine similarity between semantic representations of question $\mathbf{e}_q$ and each memory utterance $\mathbf{e}_u$: $s_u^{(\text{rel})} = sim\left(\mathbf{e}_q, \mathbf{e}_u\right)$, where $\mathbf{e}_q$ and $\mathbf{e}_u$ are semantic embedding vectors obtained from an embedding model for the query and utterance, respectively.

**Importance** Unlike querying specific details, users frequently ask for recommendations where relevance-only retrieval fails to return appropriate memories, as seen in Figure 1. To capture the

importance of utterances within the large-scale memory records, we apply graph mining on the *associative memory graph*. Drawing inspiration from PageRank in web search (Page et al., 1999), we apply Personalized PageRank (PPR) (Wayama & Sugiyama, 2025) to decide the importance of each clue and memory utterance w.r.t a given query.

$$\mathbf{r}^{(k+1)} = dM\mathbf{r}^{(k)} + (1-d)\mathbf{t} \tag{2}$$

where $M \in \{0,1\}^{N \times N}$ is the adjacency matrix from graph connectivity, $\mathbf{t} \in \mathbb{R}^{N \times 1}$ is the personalized teleportation vector, $\mathbf{r}$ is the pagerank score vector and $d$ is the damping factor. In our setting, the utterance cells in $\mathbf{t}$ are set to the similarity between query and utterance, and the clue cells are set to 0. We initialize $\mathbf{r}^0 = \mathbf{t}$. The importance score for $u$ is $s_u^{(\text{imp})} = r_u$ after convergence. Notably, we apply PPR rather than global pagerank ($\mathbf{r}^0 = \{1/N\}$) to avoid boosting the importance of memories irrelevant to the question.

**Temporal match** Temporal questions are common in the real world yet relevance cannot effectively capture the temporal constraints, as illustrated in the example in Section 1. *Recency decay*, commonly used in temporal memory retrieval, does not satisfy explicitly specified temporal constraints (Li et al., 2023). We thus conduct explicit temporal match in three steps. First, we extract temporal tokens from the question to determine if the temporal reasoning is needed. Second, we apply temporal embedding (i.e., TimeLlaMa (Yuan et al., 2024)) on the extracted temporal tokens. Third, we compute similarity between the temporal embeddings of $q$ and the utterance $u$. Here, the temporal embeddings of utterances are computed based on the temporal tokens and the timestamp associated: $s_u^{(\text{temp})} = sim\left(\mathbf{e}_q^{(\text{temp})}, \mathbf{e}_u^{(\text{temp})}\right)$

**Mutual Information for Score Fusion**[1] To balance the information from each dimension with respect to the query type, we need a strategy that can perceive the importance of each signal. *Mutual Information (MI)* is known for its ability of representing the informativeness of two variables, providing a solution to adaptively assign weights for different dimensions w.r.t query types. In our scenario, we use *Conditional MI (CMI)* to indicate *how well a signal from each dimension reflects the likelihood that a memory utterance is useful for answering a given question*.

Initially, the raw score $\tilde{s}^{(d)_u}$ of each memory $u$ for dimension $d$ is converted into three bins: *low*, *medium*, *high*. By doing so, we collect the score-label pairs $(\tilde{s}_u^{(d)(b)}, y_m^\lambda)$, where $b \in (low, medium, high)$ denotes the score bin, $\lambda \in \Lambda = \{0,1\}$ represents the memory usefulness label. The probabilities $p(\tilde{s}_u^{(d)(b)}, y_m^\lambda)$, $p(\tilde{s}_u^{(d)(b)}, q)$, $p(y_m^\lambda, q)$ can be calculated based on the collected pairs. For each query type $q$, we compute the conditional mutual information:

$$CMI_d(q) = I(\tilde{s}_u^{(d)(b)}; \lambda \mid q) = \sum_{\tilde{s}_u^{(d)(b)}} \sum_\lambda p(\tilde{s}_u^{(d)(b)}, y_m^\lambda) \log \frac{p(\tilde{s}_u^{(d)(b)}, y_m^\lambda \mid q)}{p(\tilde{s}_u^{(d)(b)} \mid q)p(y_m^\lambda \mid q)} \tag{3}$$

where $\lambda$ is the usefulness label. The weight for each dimension is then: $w^{(d)}(q) = \frac{\exp(CMI_d(q)/T)}{\sum_{d'} \exp(CMI_{d'}(q)/T)}$. The final score for each memory item $u$ is:

$$\text{Score}(q, u) = w^{(\text{rel})}(q)\, \tilde{s}_u^{(\text{rel})} + w^{(\text{imp})}(q)\, \tilde{s}_u^{(\text{imp})} + w^{(\text{temp})}(q)\, \tilde{s}_u^{(\text{temp})} \tag{4}$$

This adaptive fusion mechanism dynamically adjusts the contribution of each dimension based on its relative impact on the current query. A temperature parameter $T$ modulates the sharpness of both the score and weight distributions, enabling smoother or more selective fusion as needed.

## 3.3 MODEL FINE-TUNING

Recent advances have highlighted a persistent gap between retrieval-based recall and generative performance in large language models (LLMs) (Yang et al., 2024; Ouyang et al., 2024). This discrepancy is often attributed to the presence of irrelevant or noisy content among the top-$K$ retrieved candidates. To address this challenge and fully leverage retrieved contextual evidence, we adopt a novel fine-tuning approach using augmented datasets informed by targeted negative and positive sampling: $\text{LLM}^* = \text{FineTune}(\text{LLM}, \mathcal{D}_{\text{QA+Mem}})$.

---

[1] we provide comparisons for different fusion strategies in appendix B.5.

**Denoising QA Dataset** $\mathcal{D}_{\text{QA+Mem}}$ denotes our QA dataset comprising queries, reference answers, and memory context. Specifically, two sampling strategies are adopted for denoising fine-tuning: (1) mixed positive and negative memory contexts to encourage evidence discrimination, and (2) negative-only contexts to improve robustness by preventing over-reliance on supporting evidence.

**Multi-task Fine-tuning.** Answering a question involves recognizing its type to better utilize the memory context and generate answers. We jointly train two tasks—*question type prediction* and *answer generation*—to capture this process. The model receives an instruction, question, and sampled memories as input, and outputs both the predicted question type and the generated answer. This setup promotes effective memory utilization conditioned on question intent.

AssoMem stands out by a novel associative memory structure and a multi-dimensional scoring system, jointly enhancing retrieval quality and generation robustness across diverse scenarios.

## 4 EXPERIMENTAL SETTINGS

### 4.1 DATASET

To evaluate AssoMem, the dataset must provide question-answer (QA) pairs and large-scale memory annotated with usefulness labels to assess both retrieval and generation quality. We adopt Long-MemEval (Wu et al., 2024a), which meets these criteria and includes six question types covering common real-world scenarios. We use the original small (s) and medium (m) subsets, and additionally construct a large (l) variant to assess robustness under increasing memory size.

To further test generalizability and promote research in memory recall, we introduce MEETINGQA, a synthetic dataset simulating real-world meetings with multi-speaker dialogues on specific topics. It provides QA pairs and historical meeting transcripts as memory, with annotated usefulness labels.

All datasets follow a consistent structure: a QA pair, a base memory consisting of historical conversations where usefulness labels are available. More dataset details are in Appendix A.

### 4.2 BASELINES & EVALUATION METRICS

To assess the effectiveness of AssoMem on conversational memory recall QA, we compare it against a suite of representative baselines covering various retrieval granularities. At the **utterance level**, we include flat utterance retrieval and Long/Short-Term Memory (LST Memory) (Zhang et al., 2024). At the **session level**, we consider flat session retrieval. Under the **multi-granularity** paradigm, we compare against session retrieval first then utterance retrieval (session-utterance), session summary and utterance retrieval (session summary mem), topic grouping (Tan et al., 2025), and MemGAS (Xu et al., 2025a). More details about baselines are in Appendix C.

We adopt standard evaluation tasks: **Memory Recall** and **Answer Generation**. For memory recall, we report Recall@k and nDCG@k; for retrieval generation, we use LLM-as-a-Judge accuracy (Acc@6, Acc@10), BERTScore, and Faithfulness (Zhang et al., 2024; Lattimer et al., 2023). All retrieval metrics are computed at the utterance level.

### 4.3 IMPLEMENTATION

For base models, we select a range of models of different sizes: LlaMA-3.3-3B-Instruct, LlaMA-3.3-70B-Instruct (Grattafiori et al., 2024), Qwen2.5-32B (Qwen et al., 2025), gpt-oss-120B (Agarwal et al., 2025). Notably, for large size models such as 70B and 120B, we do not fine-tune while for other models we follow section 3.3 to perform fine-tuning via trl (von Werra et al., 2020) and transformers (Wolf et al., 2020) libraries to reduce the noise of generation. For retrievers, we select three state-of-the-art embedding models[2]: DragonPlus, DragonPlusRoberta(Lin et al., 2023), BGE(Chen et al., 2023). For temporal dynamics, we use TimeLlaMA (Yuan et al., 2024) as the embedding model. For graph construction and usage, we use networkx (Hagberg et al., 2008).

---

[2]All the retrieval results are using DPR as retriever, retriever comparisons and analysis are in Appendix B.4.

Table 1: Retrieval and QA performance on **LongMemEval** medium $m$ (the top table), large $l$ (the middle table), **MeetingQA** (the bottom table).

| Method | R@1 | R@3 | R@6 | R@10 | nDCG@3 | nDCG@6 | nDCG@10 | Acc@6 |
|---|---|---|---|---|---|---|---|---|
| *Utterance-level: retrieve top-k utterances as context* | | | | | | | | |
| Utterance-flat | 46.45 | 54.90 | 64.25 | 70.18 | 56.24 | 66.14 | 68.04 | 48.66 |
| LST Memory | 52.91 | 58.03 | 65.91 | 70.69 | 59.82 | 67.34 | 68.68 | 50.96 |
| *Session-level: retrieve top-k sessions as context* | | | | | | | | |
| Session-flat | 56.91 | 64.81 | 70.86 | 78.93 | 67.19 | 71.68 | 76.37 | 51.93 |
| *Hybrid: first retrieve sessions/topics/summaries/clues, then retrieve utterances as context* | | | | | | | | |
| Session summary mem. | 50.19 | 59.93 | 62.95 | 72.84 | 61.68 | 64.25 | 68.79 | 49.35 |
| MemGAS | 45.75 | 51.06 | 66.93 | 77.02 | 53.59 | 66.97 | 69.46 | 51.23 |
| Session-utterance | 55.37 | 64.66 | 70.17 | 78.97 | 66.04 | 71.31 | 76.50 | 55.85 |
| Topic grouping | 55.98 | 65.91 | 76.47 | 79.14 | 66.12 | 77.03 | 78.86 | 59.95 |
| **AssoMem** | **59.73** | **72.96** | **80.87** | **84.96** | **75.36** | **81.30** | **82.93** | **64.01** |
| **Method** | **R@1** | **R@3** | **R@6** | **R@10** | **nDCG@3** | **nDCG@6** | **nDCG@10** | **Acc@6** |
| *Utterance-level: retrieve top-k utterances as context* | | | | | | | | |
| Utterance-flat | 37.61 | 41.66 | 49.91 | 54.56 | 42.83 | 49.94 | 53.68 | 41.36 |
| LST Memory | 36.19 | 41.93 | 51.49 | 56.16 | 43.06 | 51.66 | 55.93 | 42.32 |
| *Session-level: retrieve top-k sessions as context* | | | | | | | | |
| Session-flat | 40.82 | 50.20 | 55.99 | 59.34 | 51.19 | 56.93 | 58.87 | 40.82 |
| *Hybrid: first retrieve sessions/topics/summaries/clues, then retrieve utterances as context* | | | | | | | | |
| Session summary mem. | 39.92 | 43.26 | 51.18 | 55.29 | 44.51 | 51.93 | 54.81 | 41.89 |
| MemGAS | 37.64 | 44.92 | 54.12 | 57.27 | 46.82 | 54.67 | 56.61 | 43.33 |
| Session-utterance | 40.93 | 49.87 | 55.83 | 59.62 | 51.92 | 56.17 | 58.93 | 45.38 |
| Topic grouping | 40.13 | 50.78 | 58.54 | 62.29 | 52.27 | 59.46 | 61.92 | 48.36 |
| **AssoMem** | **43.56** | **59.60** | **64.93** | **69.33** | **62.61** | **65.87** | **66.31** | **52.59** |
| **Method** | **R@1** | **R@3** | **R@6** | **R@10** | **nDCG@3** | **nDCG@6** | **nDCG@10** | **Acc@6** |
| *Utterance-level: retrieve top-k utterances as context* | | | | | | | | |
| Utterance-flat | 23.89 | 40.27 | 48.81 | 55.29 | 41.91 | 53.83 | 56.19 | 45.91 |
| LST Memory | 23.93 | 42.77 | 53.62 | 58.83 | 43.96 | 55.78 | 59.17 | 48.36 |
| *Session-level: retrieve top-k sessions as context* | | | | | | | | |
| Session-flat | 28.62 | 47.36 | 56.72 | 60.19 | 48.91 | 59.31 | 62.97 | 51.79 |
| *Hybrid: first retrieve sessions/topics/summaries/clues, then retrieve utterances as context* | | | | | | | | |
| Session summary mem. | 26.56 | 45.25 | 52.66 | 54.17 | 47.83 | 53.19 | 55.93 | 47.87 |
| Session-utterance | 33.78 | 52.19 | 67.63 | 77.91 | 55.31 | 69.48 | 80.19 | 49.17 |
| MemGAS | 32.94 | 52.59 | 69.34 | 80.67 | 55.86 | 71.66 | 82.93 | 61.26 |
| Topic grouping | 39.66 | 61.69 | 78.98 | 89.15 | 63.15 | 74.23 | 83.78 | 63.56 |
| **AssoMem** | **41.63** | **64.72** | **85.17** | **92.96** | **66.06** | **86.93** | **94.17** | **69.41** |

## 5 EXPERIMENTAL RESULTS

In this section, we will present the thorough experimental results and concrete analysis with a focus on answering these research questions, supplementary results can be seen in Appendix B:

- **RQ 1.** Does AssoMem exhibit performance advantages over state-of-the-art solutions?
- **RQ 2.** How does RITRANKER contribute to performance of AssoMem?
- **RQ 3.** How robust is AssoMem against the memory size and question types?

We will answer RQ 1. in Section 5.1, RQ 2. in Section 5.2 and RQ 3. in Section 5.3 and 5.4.

### 5.1 COMPARATIVE STUDY

**Retrieval Results.** As presented in Table 1[3], on LongMemEval_m dataset, the Session-utterance retrieval raises R@10 to 78.97% and nDCG@10 to 76.50% over the utterance-level flat retrieval

---

[3]We adopt 6 as a reference point based on the retrieval–generation transition analysis in Appendix B.3.

Table 2: Generation results of different LLMs using AssoMem recall@10 retrieval as context.

| Model | Accuracy | | BertScore | | Faithfulness | |
|---|---|---|---|---|---|---|
| | Plain | Fine-tuned | Plain | Fine-tuned | Plain | Fine-tuned |
| LlaMA3.2-3B-Instruct | 26.91 | 33.43 | 31.19 | 36.93 | 33.27 | 43.06 |
| Qwen2.5-32B-Instruct | 64.72 | 73.88 | 67.91 | 77.49 | 55.16 | 75.73 |
| LlaMA-3.3-70B-Instruct | 65.83 | - | 71.86 | - | 56.96 | - |
| Gpt-Oss-120B | 76.49 | - | 81.76 | - | 68.73 | - |

at 70.18% and 68.04%. Topic grouping further improves R@10 to 79.14% and nDCG@10 to 78.86%. Similar patterns are detected on LongMemEval_l and MeetingQA datasets. This suggests a general finding that hybrid retrieval clearly outperforms single-granularity retrieval. Our proposed AssoMem, on the other hand, makes an improvement of 5.82% over the SOTA, topic grouping. Further, AssoMem also makes improvements of 7.04% and 3.81% against the best baselines on LongMemEval_l and MeetingQA, respectively. This suggests that *AssoMem does offer advantages compared with SOTA. Why do prior memory methods fail at scale?* As memory size grows, more similar memory records accumulate where similarity alone cannot discriminate among many near-duplicate or thematically close candidates, so recall collapses and downstream generation suffers. *Why does AssoMem help?* By ranking with importance and temporal priors in addition to relevance, AssoMem takes the question types into consideration and performs retrieval from a multi-dimensional anchor. As can be seen in Figure 3(a), we clearly witness the improvements made in preference and temporal reasoning type questions which suggests the success of our scoring system.

**Generation Results** Retrieval quality translates directly to generation: Acc@6 climbs from 48.66 for flat retrieval to 55.85% for multi-granularity and to 64.01% for AssoMem, with BERTScore moving from 51.71% to 60.06% and then to 67.56%. This again validates that AssoMem is making improvements against SOTA. Moreover, as previous studies have verified that there exists a gap between the recall and generation performance Yang et al. (2024); Ouyang et al. (2024) which means the top-k retrieved content may also deliver noise into the context for downstream question answering (We present analysis in Appendix B.3 to support this claim). The results presented in Table 1 also validate this where the recall@6 result of AssoMem is 80.87% while the answering accuracy@6 is 64.01 on m dataset, a similar pattern is also observed on l dataset. Thus, the base model's ability to fully utilize the retrieved memory can be a key to improve the downstream generation performance. Following the fine-tuning strategy in Section 3.3, the generation accuracy@10 gains improvements of 6.52% and 9.16% for LlaMA3.2-3B and Qwen2.5-32B models, respectively as in Table 2. This verifies that our fine-tuning strategy helps the model better utilize the retrieved memory context, contributing to the AssoMem performance.

## 5.2 ABLATION STUDY

**Ablation on retrieval dimensions** As can be seen in Figure 3 (b), the performance on temporal reasoning type questions drops when excluding temporal dimension information and the performance on single-user-preference type questions drops when excluding importance dimension information. These observations validate the challenge mentioned in Section 1 that different types of questions require information from different dimensions for retrieval, illustrating that *RITRanker* contributes to AssoMem's performance by well integrating signals from multiple dimensions.

**Ablation on components** In AssoMem, the associative memory graph serves as the base for obtaining multi-dimensional information while the MI-guided weight assignment strategy serves as their connections. Thus, we further tested how each component impacts the performance. Specifically, we conduct experiments under these two settings: 1. remove the clue nodes within the graph; 2. remove the weight assignment strategy and instead, use a fixed weighted sum for comparisons. As can be observed in Table 3, the

Table 3: Ablation on components and dimensions. w/o denotes without the component compared to AssoMem.

| w/o | R@6 | R@10 | Acc@6 | Acc@10 |
|---|---|---|---|---|
| Temporal | 73.39 | 78.37 | 57.88 | 61.19 |
| Importance | 75.81 | 79.62 | 59.55 | 61.97 |
| Clue nodes | 79.75 | 84.80 | 63.06 | 72.51 |
| Weight Assignment | 76.79 | 81.80 | 60.38 | 58.89 |
| AssoMem | **80.87** | **84.96** | **64.01** | **69.17** |

retrieval performance drops 1.12% without clue nodes which we attribute to the fact that the clue level retrieval would be suboptimal when the importance information is missing. Moreover, using a fixed weight assignment results in a 4.08% performance drop compared to full AssoMem which further validates the necessity of each component within AssoMem[4].

## 5.3 ROBUSTNESS STUDY

**Results on different question types.** As in Figure 3 (a), we can clearly observe that the perfor-

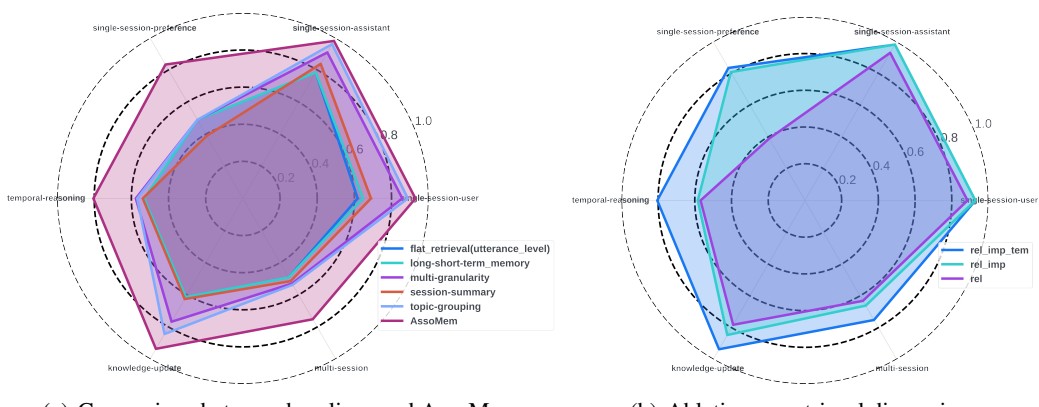

(a) Comparison between baselines and AssoMem      (b) Ablation on retrieval dimensions

Figure 3: The radar figure showing performance on different question types.

mance of AssoMem , beyond end-to-end performance in Table 1, consistently outperforms all baselines across six question types. Notably, AssoMem presents the largest margins on preference and temporal type questions compared with other baselines which we attribute to the fact that similarity-only retrieval over-emphasizes surface relevance and tends to return most similar memories, which is adequate for pure recall but brittle for preference, temporal, and cross-session questions.

**Results on memory size** As presented in Table 1, AssoMem makes improvements of 6.39%, 7.04% for the R@6, R@10 against the best baseline - topic grouping, respectively. Following the retrieval performance improvements, the generation accuracy performance is also improved by 4.06%. Together with the advantages made on LongMemEval_m, the results suggest that our AssoMem present a strong robustness as the memory size increases since from m to l, the dialogue sessions increase from 500 to 2,500 rounds per data point. These results and observations on increased memory size and the diverse question types highlight the strong robustness of AssoMem.

## 5.4 ERROR ANALYSIS

As shown in Table 4, AssoMem achieves the highest retrieval fidelity, with a Correct rate of 64.01%—exceeding Topic Grouping by 4.06% and LST Memory by 13.05%—and the lowest Retrieval Error at 19.13%, reducing errors by 4.30% and 14.96% respectively. It also yields the lowest Wrong-grounding rate (2.86% vs. 3.82% and 5.13%), indicating more accurate and intent-aligned context. However, better retrieval does not fully translate into generation quality: total generation errors remain compa-

| Error bucket | LST Mem. | Topic | AssoMem |
|---|---|---|---|
| Correct | 50.96 | 59.95 | 64.01 |
| RE: Incorrect retrieval | 34.09 | 23.43 | 19.13 |
| GE: Wrong grounding | 5.13 | 3.82 | 2.86 |
| GE: Misuse of positives | 2.22 | 4.37 | 5.71 |
| GE: LLM-Judge error | 7.60 | 8.43 | 8.29 |

Table 4: Error analysis. RE denotes retrieval error, GE denotes generation error. Wrong grounding means negatives are used for generation, misuse of positives means the positives are misused by LLM.

rable at 16.86% for AssoMem, and 16.62% for Topic Grouping. Overall, AssoMem's advantage lies

---

[4]Comparisons on different fusion strategies are provided in B.8 and B.5

in reducing retrieval-side failures and confusing negatives; the remaining gap is generation-side, suggesting the need for stronger evidence utilization which necessitates our fine-tuning strategy. These observations again validate the effectiveness and robustness of proposed AssoMem.

## 5.5 LATENCY & COST STUDY

Beyond retrieval and generation quality, practical deployment of memory QA systems demands low latency and efficient token usage. We benchmark the end-to-end query latency and token consumption of AssoMem against recent graph-based baselines on LongMemEval_m (Table 5).

**Query-time efficiency.** AssoMem achieves an average query latency of 1.30 s, which is 3.5× lower than HippoRAG (4.51 s) and 1.9× lower than

Table 5: Latency and token cost on **LongMemEval_m**. *Tokens* = average input tokens consumed for generation per query.

| Method | R@3 | R@6 | Acc@6 | Tokens | Lat. (s) |
|---|---|---|---|---|---|
| HippoRAG | 49.66 | 64.18 | 48.61 | 8,430 | 4.51 |
| A-Mem | 47.82 | 62.17 | 48.06 | 8,852 | 2.45 |
| **AssoMem** | **72.96** | **80.87** | **64.01** | **1,846** | **1.30** |

A-Mem (2.45 s). This speedup stems from two factors: (i) AssoMem retrieves at the utterance level without requiring an expensive LLM-based Open Information Extraction step at query time, and (ii) the RITRanker scoring pipeline—comprising PPR (0.21 s), weight assignment (0.26 s), and answer generation (0.74 s)—is lightweight by the structure design[5].

**Token efficiency.** AssoMem consumes only 1,846 tokens on average per query, roughly 4.6× fewer than HippoRAG (8,430) and A-Mem (8,852). Because HippoRAG and A-Mem construct fine-grained entity-level graphs, their retrieval pipelines surface a larger volume of fragmented evidence that must be concatenated into the prompt, inflating token cost without proportional accuracy gains.

**Offline construction cost.** Full graph construction for 2,500 sessions (24.4k utterances) takes approximately 1,950 s, which is 23% less than MemGAS (2,398 s). More importantly, incremental updates—the dominant mode in production—require only 0.01 s per new node, making AssoMem practical for continuously growing memory stores.

## 6 CONCLUSION & FUTURE WORK

In this work, we addressed the critical challenge of accurate, scalable memory recall in conversational AI by tackling the limitations of relevance-only retrieval in large-scale memory scenarios. We proposed AssoMem, a novel framework that enhances retrieval quality and generation robustness across diverse query types. At its core, AssoMem constructs memories as an associative graph and employs the RITRanker system to align relevance, importance, and temporal retrieval signals. To support future research, we introduce MEETINGQA, a synthetic multi-speaker dataset simulating real-world meeting scenarios, with annotated QA pairs and memory usefulness labels. Extensive experiments across all datasets demonstrate the effectiveness and robustness of AssoMem.

Building upon the success of AssoMem, future research will focus on the following extensions: 1. Extending the framework to manage memory settings that involve the accumulation of heterogeneous and evolving histories sourced from multiple modalities; 2. Expanding the associative memory clues by incorporating richer semantic concepts such as events, locations, and external knowledge bases; 3. Developing personalized memory compression techniques to facilitate efficient on-device deployment of the associative memory system.

---

[5]We further provide comprehensive latency analysis in Appendix B.7

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

# A    APPENDIX. DATASETS DETAILS

Table 6: Corpus statistics for **LongMemEval** and **MeetingQA**.

| Dataset | Averages | | | | | Haystack Totals | |
|---|---|---|---|---|---|---|---|
| | #Sessions | Evid./Q | Q tokens | A tokens | Utts./sess. | Session tokens | Mem Tokens (m) |
| *LongMemEval_s* | 50.2 | 1.9 | 17.3 | 9.9 | 9.83 | 2053 | 0.10 |
| *LongMemEval_m* | 501.9 | 1.9 | 17.3 | 9.9 | 9.76 | 2016 | 1.01 |
| *LongMemEval_l* | 2503.4 | 1.9 | 17.3 | 9.9 | 9.76 | 2016 | 5.13 |
| *MeetingQA* | 50 | 1 | 10.17 | 2.18 | 72.32 | 1848.62 | 0.09 |

"Evid./Q" = average number of answer evidences per question; "Utts./sess." = average utterances per session; m stands for million.

As detailed in Table 6, we evaluate all the baselines and proposed AssoMem across four datasets. Among them, the *LongMemEval_s* and *LongMemEval_m* are open benchmarks. We enlarge this benchmark into a larger scale by incorporating more dialog sessions into the memory and form the *LongMemEval_l* which consists of 2,500 dialog sessions. Notably, the added dialog sessions are identical, ensuring the dataset quality.

Further, we mimic the real-world meeting scenario where different speakers will speak during a meeting, yielding a meeting session with different turns, and construct a new dataset named MeetingQA. The purpose of this dataset is for evaluating memory recall question answering in multi-turn dialog scenario. The MeetingQA dataset is a collection of 50 meetings, each containing approximately 70–80 messages, with a variety of speaker configurations (ranging from 2 to 6 speakers per memo). The dataset was generated by a linguistic engineer using LLMs and is designed to support benchmarking and development of meeting recall systems. It includes 390 QA pairs, where each answer is mapped to at least one specific message within a corresponding meeting. The dataset structure provides detailed fields such as message IDs, session IDs, speaker identifiers (typically generic unless manually annotated), and transcript messages.

In tandem, the data we used in the experiments follows this structure: {question: ..., answer: ..., question_date: ..., sessions: { { session_id: 1, utterance: ..., date: ...}, { session_id: 2, utterance: ..., date: ...}, ...} }. We organize the dialog memories and retrieve utterances to answer the question.

Table 7: Question types and required retrieval dimensions

| Question | R | P | T |
|---|---|---|---|
| Where did I have dinner yesterday? | ✓ | | ✓ |
| What do I usually say at work? | | ✓ | |
| Most visited coffee shop last month? | | ✓ | ✓ |
| How long have been since I start my job? | ✓ | | ✓ |

Table 8: Retrieval and QA performance on **LongMemEval** small $s$.

| Method | R@1 | R@3 | R@6 | R@10 | nDCG@6 | nDCG@10 | Acc@6 | BERTScore |
|---|---|---|---|---|---|---|---|---|
| *Utterance-level: retrieve top-k utterances and return utterances as context* | | | | | | | | |
| Utterance-level | 52.43 | 65.55 | 79.63 | 93.19 | 75.37 | 83.23 | 59.13 | 68.61 |
| LST Memory | 49.20 | 63.93 | 82.98 | 93.50 | 79.77 | 83.66 | 62.17 | 74.39 |
| Embedding_cat | 50.13 | 60.90 | 76.25 | 86.17 | 71.93 | 80.21 | 55.96 | 58.06 |
| *Session-level: retrieve top-k sessions and return sessions as context$^{\dagger}$* | | | | | | | | |
| Session-level | 57.04 | 66.19 | 81.63 | 94.96 | 78.15 | 85.72 | 53.67 | 56.17 |
| *Hybrid: first retrieve topic/session, then utterances within; return utterances as context* | | | | | | | | |
| Multi-granularity | 59.92 | 76.73 | 83.89 | 94.08 | 82.59 | 85.61 | 63.74 | 73.61 |
| Session summary mem. | 58.21 | 77.06 | 79.96 | 85.17 | 77.18 | 79.48 | 60.37 | 70.73 |
| Topic grouping mem. | 61.84 | 81.73 | 90.07 | 96.31 | 88.63 | 91.47 | 68.11 | 78.91 |
| **AssoMem** | **62.73** | **81.96** | **93.87** | **96.96** | **90.31** | **91.93** | **72.13** | **80.65** |

# B  APPENDIX. SUPPLEMENTARY RESULTS

## B.1  RESULTS ON LONGMEMEVAL_S

Table 8 corroborates the main-text trends under the small-$s$ setting. Pure utterance retrieval remains weakest, while session-only retrieval improves recall but limits QA due to context dilution. Hybrid pipelines are consistently stronger: multi-granularity reaches 78.97 R@10 and 76.50 nDCG@10 with 55.85 Acc@6, and topic grouping further lifts retrieval to 79.14 R@10 and 78.86 nDCG@10 with 59.95 Acc@6. **AssoMem** attains the best results across all metrics—84.96 R@10 and 82.93 nDCG@10—translating into 64.01 Acc@6 and 67.56 BERTScore, which improves over the utterance baseline by 14.78 R@10, 14.89 nDCG@10, 15.35 Acc@6, and 15.85 BERTScore; it also surpasses topic grouping by 5.82 R@10 and 4.06 Acc@6. The persistence of these gains at smaller $s$ indicates that popularity- and temporal-aware re-ranking continues to suppress stale or idiosyncratic items and surfaces recent, widely supported evidence, yielding both higher retrieval concentration and better downstream generation. The results presented in Table 8 consistently validate our findings as in the main text.

## B.2  WHY PPR?

To better understand how importance impact on the performance, we further provide the result comparisons between personalized pagerank and pagerank. As presented in Figure 4 (a), we witnessed that the ppr consistently outperforms the pr in single-session-preference type question which can be attributed to the fact that answering preference question requires both the relevance for locating the event and the importance for knowing the most important memories for the located event. In this sense, ppr provides better importance due to the fact that ppr considers relevance in both the algorithm initialization and teleportation.

## B.3  PERFORMANCE GAP BETWEEN RETRIEVAL AND GENERATION

Across top-k, retrieval monotonically improves for all methods, with AssoMem leading at every k and saturating near 0.85 by k=10, ahead of Topic Grouping and Multi-granularity that level off around 0.80 and 0.79. QA accuracy, however, plateaus much earlier: AssoMem rises quickly to about 0.66 by k=6 and then gains marginally; Topic Grouping continues to climb and ends highest near 0.80, while Multi-granularity trails and tops out near 0.60. The widening retrieval–accuracy

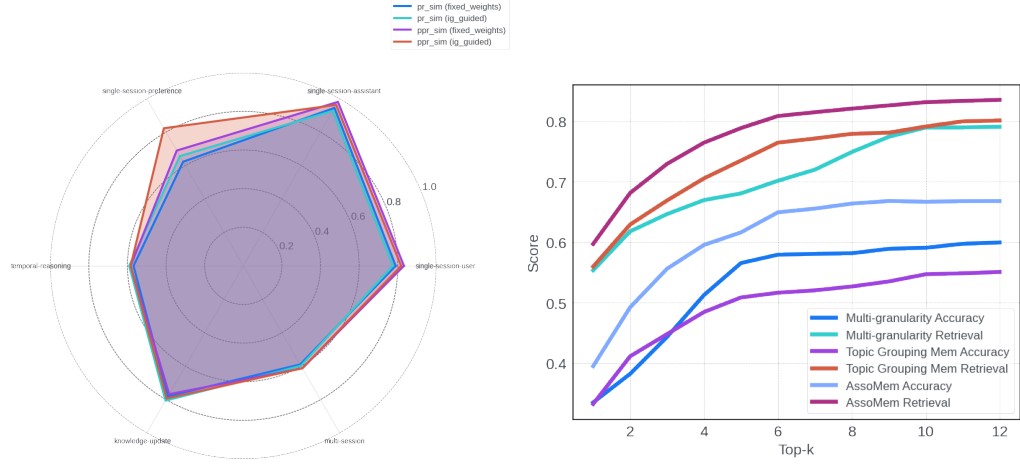

(a) PPR vs. PR on different graph structure

(b) Recall vs. Generation of representative baselines and AssoMem

Figure 4: Supplementary results

| Granularity | Retriever | LongMemEval_s | LongMemEval_m | LongMemEval_l | MeetingQA |
|---|---|---|---|---|---|
| Utterance | DP | 72.54 | 58.31 | 44.62 | 42.94 |
| | BGE | 78.89 | 64.18 | 48.73 | 48.27 |
| | DPR | **79.63** | **64.25** | **49.91** | **48.81** |
| Session | DP | 77.26 | 64.81 | 50.33 | 50.79 |
| | DPR | 81.63 | 70.86 | 55.99 | **56.72** |
| | BGE | **83.79** | **72.17** | **56.43** | 55.92 |

Table 9: Recall@6 performance of three retrievers: DragonPlus, DragonPlus-Roberta and BGE (Large) across 4 datasets in two granularity levels.

gap beyond when k is 6 indicates diminishing returns from adding more context and highlights a utilization bottleneck: even with superior retrieval, the LLM cannot fully exploit larger evidence sets due to redundancy, distraction, or context-window limits. Practically, this suggests a sweet spot around k=6 and the need for post-retrieval evidence organization (summarization, attribution, or re-weighting) to convert AssoMem's retrieval gains into further accuracy improvements.

## B.4 RETRIEVER COMPARISONS

Retrievers play a pivotal role in the modern memory systems, providing semantic relevance information for locating useful memory records. As presented in Table 9, we can witness that in general, DragonPlus shows sub-optimal performance compared with other two retrievers across 4 datasets in two granularities. Furthermore, DragonPlusRoberta consistently outperforms other two retrievers by 7.09%/0.74%, 5.94%/0.07%, 5.29%/1.18% and 5.87%/0.54% across 4 datasets in utterance-level retrieval, respectively. This indicates a comparable performance between DPR and BGE (Large). Moreover, in session level retrieval, BGE (Large) on the opposite, consistently outperforms other two retrievers by 6.53%/2.16%, 7.36%/1.31% and 6.1%/0.44% across three LongMemEval datasets, respectively. We attribute this to the fact that the larger context window of BGE (large) helps boost the retrieval performance in session level since the context is longer. However, in session-level retrieval on MeetingQA, the performance of BGE (large) is suboptimal compared to DPR, this is because the token length per session in MeetingQA is not comparable with LongMemEval and can be handled by DPR. In tandem, we can conclude that within the capability of DP model, DPR is better than BGE (Large) and since our most fine-grained granularity in this work is utterance-level retrieval we opt for DPR as the main retrieving embedding model.

| Question Type | Ground-Truth Memory Evidence | Top-2 Retrieval (Relevance-Only) | Top-2 Retrieval (AssoMem) |
|---|---|---|---|
| **Q1: Preference reasoning** *"What do I usually complain at work, can you give some tips to avoid it?"* | *We always start planning too late, it's like we're just reacting.* *Every time we plan, there seems no room for unexpected stuff.* (High-importance opinions repeated across history.) | **Error:** *I worked late few nights this week, working on the new API integration.* **Error:** *The search team was complaining our work progress in the sync.* (Semantically related to "work," but not representative of core opinions.) | **Correct:** *To be honest, I think the results on project memory is too good to trust.* **Correct:** *To be honest, I think search team can help us on the project launching.* (High-importance, repeated memory retrieved.) |
| **Q2: Temporal reasoning** *"Which show did I watch first, the Crown or the Game of Throne?"* | *I just finished watching the third season of 'The Crown' on Netflix.* *I started watching 'the Game of Throne' about one month ago.* (Provides necessary temporal anchors.) | **Error:** *Do you have similar show suggestions such as 'the Crown', 'the Game of Throne'?* **Error:** *"Game of Throne" is an epic show, I'm pretty sure you're hooked. If you're looking for some similar shows to "Game of Throne", I have some recommendations.* (Topically related but lacks any temporal signal.) | **Correct:** *I started watching 'the Game of Throne' about one month ago.* **Correct:** *I just finished watching the third season of 'The Crown' on Netflix.* (Accurately supports temporal comparison.) |

Table 10: Case study showing limitations of relevance-only retrieval across question types. While relevance methods retrieve semantically similar content, they often miss task-relevant evidence. AssoMem incorporates importance and temporal signals to enable more accurate memory selection.

## B.5 ASSOMEM WITH DIFFERENT FUSION STRATEGIES

Further, we compare 6 information strategy from two types: Information theory driven weight assignment: Information Gain (Datta et al., 2022), Mutual Information (Zhang et al., 2023); Learnable weight assignment (Shah et al., 2020): Logistic Regression (LR), Random Forest (RF), Support Vector Machine (SVM), Two-layer linear network (LN).

**Impact of Different Information Fusion Strategy** We further present the results of comparing different weight assignment strategies as in Table 11. We can observe the performance of information driven weight assignment strategies in general outperforms the learnable weight assign-

Table 11: AssoMem with different fusion strategies.

| Fusion strategy | Recall@6 | Recall@10 | Acc@6 |
|---|---|---|---|
| *Learnable weight assignment* | | | |
| Logistic Regression | 72.92 | 78.55 | 58.43 |
| Random Forest | 77.92 | 80.69 | 61.71 |
| Linear Network | 77.29 | 80.63 | 61.74 |
| Support Vector Machine | 78.81 | 81.66 | 62.11 |
| *Information driven weight assignment* | | | |
| Information gain | 80.94 | 86.33 | 63.84 |
| Mutual information | **82.64** | **88.91** | **64.20** |

ment strategies which we attribute this to the fact in the conversational memory recall question answering scenario, the memory evidence for each question is sparse which poses difficulties for simply training a learnable model to do weight assignments. On the other hand, information driven strategies use the information purity as the signal which mitigates the memory sparsity influence. Another benefit for using information driven strategy is training-free, we see a great potential of learnable strategies to handle weight assignment yet the trade-off might also be a factor that hinders its application in this scenario.

Table 12: Overview of baseline methods by granularity

| Granularity | Method | Description |
|---|---|---|
| Utterance level | Long- and short-term mem. (Zhang et al., 2024) | Partitions user memory into long- and short-term components to enable better coordination during memory recall QA. |
| | Flat Retrieval | Retrieves utterances directly based on relevance scores. |
| Session level | Flat Retrieval | Retrieves entire sessions based on overall relevance to the query. |
| Multi-granularity | Session-utterance | First retrieves relevant sessions (based on full session content), then retrieves relevant utterances within those sessions. |
| | Session summary-utterance | First retrieves sessions based on session summaries, followed by utterance-level retrieval within those sessions. |
| | Topic-utterance grouping (Tan et al., 2025) | Groups utterances by topic, retrieves relevant topics, then selects utterances within those retrieved topic groups. |
| | MemGAS (Xu et al., 2025a) | Uses four granularities (utterance, session, keyword, summary) with entropy-guided weighting for retrieval across these levels. |

## B.6 CASE STUDY

Table 10 presents a case study of retrieval behavior across two question types—preference reasoning and temporal reasoning—highlighting the limitations of relevance-only retrieval and the improvements achieved by AssoMem. In Q1 (Preference reasoning), the user asks: "What do I usually say at work?" This question requires retrieval of high-importance utterances that reflect consistent patterns in user opinions. However, the relevance-only method retrieves memories that are superficially related to "work" but do not reflect the user's repeated viewpoints. In contrast, AssoMem successfully identifies two high-importance memory entries expressing critical opinions about project memory—these are not only topically relevant but also semantically central to the user's past behavior. This demonstrates AssoMem's ability to incorporate an importance signal, which is essential for preference-centric questions. In Q2 (Temporal reasoning), the user asks: "Which show did I watch first, the Crown or the Game of Throne?" This question necessitates precise temporal comparison, which relevance-only methods fail to address. The retrieved responses are thematically related to the queried shows but offer no chronological cues. In contrast, AssoMem correctly surfaces two time-anchored memory entries indicating both the start time of The Game of Throne and the completion of The Crown, enabling accurate temporal reasoning. In tandem, the case study confirms that relevance alone is insufficient for diverse real-world queries. By incorporating importance and temporal dynamics, AssoMem delivers more contextually accurate retrieval, leading to better generation.

## B.7 LATENCY ANALYSIS

We report the average latency (in seconds) of each core component in the AssoMem pipeline to assess its efficiency. The most time-consuming operation is memory construction, taking approximately 1948.99 seconds on average, which occurs as a one-time offline process to build the structured memory graph. In contrast, incremental operations during inference are significantly faster. Node addition and edge addition require only 0.01 and 13.96 seconds respectively, enabling dynamic updates with low overhead. RIT scoring and weight assignment, which compute information-theoretic rele-

Table 13: AssoMem latency statistics.

| Operation | Avg. Latency (s) |
|---|---|
| Graph Construction | 1948.99 |
| Node Addition | 0.01 |
| Edge Adding | 13.96 |
| RIT Scoring | 0.39 |
| Weight Assignment | 0.26 |
| Answer Generation | 0.74 |

vance and balance multi-dimensional signals, incur negligible latency of 0.39 and 0.26 seconds. The final inference stage, which utilizes retrieved and weighted memory as context for LLM generation, completes within 0.74 seconds on average. These results demonstrate that although initial memory graph construction is computationally expensive, the online inference and memory aug-

mentation steps remain efficient, ensuring the system is practical for real-time applications with dynamic memory updates.

## B.8 CMI FUSION VS. NON-ADAPTIVE BASELINES

Table 10 (Appendix B.5) compares CMI against several learnable and information-theoretic fusion strategies. Here we further include a *non-adaptive* fixed-weight baseline to isolate the benefit of adaptive weighting. As shown in Table 14, CMI outperforms the fixed assignment by **7.7%** in Recall@6 and **8.1%** in Acc@6. The non-adaptive baseline applies a single set of weights across all query types, which is suboptimal because different question categories (e.g., temporal reasoning vs. preference) rely on different signal distributions. CMI captures these conditional dependencies between each scoring dimension and the usefulness label, yielding consistently higher retrieval and generation quality.

Table 14: Fusion strategy comparison on **LongMemEval_m**. *Non-adaptive* assigns fixed weights 0.6/0.3/0.1 to R/I/T.

| Fusion Strategy | R@6 | R@10 | Acc@6 |
|---|---|---|---|
| Non-adaptive | 76.70 | 80.80 | 59.38 |
| SVM | 78.81 | 81.66 | 62.11 |
| Information Gain | 80.94 | 86.33 | 63.94 |
| **CMI (ours)** | **82.64** | **88.91** | **64.20** |

## B.9 LONG-CONTEXT VS. RETRIEVAL-AUGMENTED APPROACHES

An alternative to retrieval-augmented generation is to feed the entire conversational history directly into a long-context LLM. We evaluate this setting on Long-MemEval_m using Llama-3.3-70B-Instruct with its full 128k context window. As reported in Table 15, the long-context approach achieves only 21.93% Acc@6, substantially below even the weakest RAG baseline (HippoRAG, 48.61%). This confirms that naively extending the context window is insufficient for similarity-dense memory

Table 15: Long-context vs. RAG on **LongMemEval_m**.

| Approach | Acc@6 |
|---|---|
| Long-Context (Llama-3.3-70B) | 21.93 |
| RAG (HippoRAG) | 48.61 |
| **AssoMem (ours)** | **64.01** |

scenarios: the model struggles to locate and attend to relevant evidence among thousands of semantically similar utterances. AssoMem's structured retrieval and multi-signal ranking effectively address this challenge, achieving **64.01%** Acc@6—a **2.9×** improvement over the long-context baseline.

## C BASELINES

The detailed descriptions are presented in Table 12.

## D APPENDIX. PROMPTS

**Generation Prompt:**

```
### TASK DESCRIPTION
You are a helpful assistant that answers user's question. In this
    sense, you
will have access to user's memory records which contain user's
    historical information.
Please note you will need to identify if the memories are useful or
    not for you to respond to the query.
If the memories are useful then answer the question based on the
    memories, otherwise answer the question based on your knowledge or
     answer "IDK".

### INPUT
User memory: {memory}
User query: {question}

### OUTPUT REQUIREMENT
```

```
        Output the answer to the question only. Not matter you use the memory
            or not, please only output the answer and nothing else.
```

**Topic Generation Prompt:**

```
    ### TASK DESCRIPTION
    You are a helpful assistant that helps users to organize their memory
        records. Next, you'll help me in organizing a user's memory
        records.
    Given a user's historical dialogue session, please summarize the
        session into a concise topic summary without key information lost.
    Output the topic summary sentence.

    ### INPUT
    Dialogue session: {session}

    ### OUTPUT REQUIREMENT
    Generate a topic summary for the given session.
    Please only output the topic summary and nothing else.
```

**LLM-as-a-Judge Prompt:**

```
    ### TASK DESCRIPTION
    You are a helpful judge to evaluate the quality of the response to a
        user question.
    You will be given a user question and two responses: one is the golden
         response, one is the generated response.
    Please evaluate the quality of the generated response based on the
        following criteria:
    a) If the response is relevant to the user question.
    b) If the response answers the question or not.
    c) If the response is consistent and coherent.
    If you think the generated response meets these criteria or
        semantically responds to the user as the golden response does, you
         should output "Win", otherwise "Lose".

    ### INPUT
    User query: {question}
    Golden Response: {response_1}
    Generated Response: {response_2}

    ### OUTPUT REQUIREMENT
    Output "Win" or "Lose" only. Do not output anything else.
```

**Fine-tuning Sample:**

```
    <|begin_of_text|><|start_header_id|>system<|end_header_id|>

    Cutting Knowledge Date: December 2023
    Today Date: 26 Aug 2025

    ### TASK DESCRIPTION
    You are a helpful assistant that answers user's questions. In this
        sense, you will have access to user's memory records which contain
         user's historical information.
    Please note you will need to identify if the memories are useful or
        not for you to answer the query.
    If the memories are useful then answer the question based on the
        memories, otherwise answer the question based on your knowledge or
         answer "IDK".

    ### OUTPUT REQUIREMENT
    Output the answer to the question only. No matter you use the memory
        or not, please only output the answer and nothing else.<|eot_id
        |><|start_header_id|>user<|end_header_id|>
```

```
### The user query is:
What is the order of the three events: 'I signed up for the rewards
    program at ShopRite', 'I used a Buy One Get One Free coupon on
    Luvs diapers at Walmart', and 'I redeemed $12 cashback for a $10
    Amazon gift card from Ibotta'?
### The memory is:
I'm planning a trip to Walmart this weekend and I'm looking for some
    deals on baby essentials. Do you have any info on their current
    sales or promotions on diapers? By the way, I used a Buy One Get
    One Free coupon on Luvs diapers at Walmart today, which was a
    great deal!;I'm planning a shopping trip to Target this weekend
    and I'm wondering if you have any info on their current sales and
    promotions. By the way, I just redeemed $12 cashback for a $10
    Amazon gift card from Ibotta today, so I'm feeling pretty good
    about my savings so far!;I'm trying to plan my grocery shopping
    trip for this week. Can you help me find any good deals or sales
    on diapers and formula at ShopRite? By the way, I signed up for
    their rewards program today, so I'm hoping to maximize my points
    and savings.<|eot_id|><|start_header_id|>assistant<|end_header_id
    |>

### The answer is:First, I used a Buy One Get One Free coupon on Luvs
    diapers at Walmart. Then, I redeemed $12 cashback for a $10 Amazon
     gift card from Ibotta. Finally, I signed up for the rewards
    program at ShopRite.<|eot_id|>
```

# E    USE OF LLM

In paper writing, we use LLMs solely for checking typos and grammar errors; they are not used for any other purposes beyond this.

