# OpenReview forum: "AssoMem: Scalable Memory QA with Multi-Signal Associative Retrieval"
_ICLR.cc/2026/Conference — ICLR 2026 Poster_

### Official Review · Reviewer_xRKF · 2025-10-18

**Soundness:** 3
**Presentation:** 3
**Contribution:** 3
**Rating:** 4
**Confidence:** 4

**Summary:**

The paper proposes AssoMem, a memory question answering framework for large, similarity-dense conversational histories. It constructs an associative memory graph that links LLM-generated “clues” to raw utterances, retrieves top clues, and then ranks candidate utterances. The ranking component, RITRanker, fuses relevance, importance estimated by query-personalized PageRank, and temporal alignment, with mutual-information guided weights that vary by question type. A multi-task denoising fine-tuning step is used to improve small models’ answer generation ability from noisy retrieved context. Experiments on LongMemEval and the new MeetingQA benchmark show consistent gains over utterance, session, and hybrid baselines.

**Strengths:**

1. This paper designs novel memory structure and ranking metrics. The method bridges the relevance-only limitation of existing memory and retrieval methods.
1. The empirical retrieval performance outperforms the baselines by a large margin. Multiple models and baselines are studied.
1. This paper contributes a new MeetingQA dataset that can be used for testing long-term memory systems in different scenarios.

**Weaknesses:**

1. HippoRAG [1,2] designs a graph-structured memory as well and also uses PPR. I would like to see some comparisons as this paper is also a graph-based method.
1. Several designs (clue generation, temporal token) are bound to chat-based tasks. By comparison, general memory architectures can be applied to and are actually tested on long-context and RAG tasks. [2, 3]
1. More details regarding the fine-tuning needs to be discussed. It seems that the fine-tuning is an optional component and does not fit with the entire memory architecture. However, it indeed affects the memory read performance a lot, according to Table 2. It is rare to specially design a fine-tuning procedure just to improve the narrow task of “reading chat memory retrieved from a graph index”. Does the fine-tuning also improve the model’s general long-context ability? Conversely, are existing long-context training recipes/models (e.g., [4] [5]) applicable to this paper?
1. I would like to see more cost analysis both in terms of latency and in terms of tokens consumed.
1. I would like to see more details and further analyses to show the value of the new dataset.

[1] Hipporag: Neurobiologically inspired long-term memory for large language models. Gutiérrez et al., 2024.

[2] From RAG to Memory: Non-Parametric Continual Learning for Large Language Models. Gutiérrez et al., 2025.

[3] M+: Extending memoryllm with scalable long-term memory. Wang et al., 2025.

[4] Make your llm fully utilize the context. An et al., 2024.

[5]  Wildlong: Synthesizing realistic long-context instruction data at scale. Li et al., 2025.


I am willing to raise my ratings if these concerns are sufficiently addressed.

**Questions:**

Please refer to the weaknesses.

---

> ### Author Response · Authors · 2025-11-22
> **Rebuttal**
>
> **In Responding to Weakness**
>
> **W1** HippoRAG [1,2] designs a graph-structured memory as well and also uses PPR. I would like to see some comparisons as this paper is also a graph-based method.
>
> **R1** The key difference between HippoRAG and our AssoMem is the granularity of the nodes. HippoRAG constructs a fine-grained graph where nodes represent entities (like in a KG), whereas nodes in AssoMem are whole utterances. The fine-grained graph maximizes the utilization of connectivity information of the graph thus suitable for tackling multi-hop questions; however, information lost in LLM-based graph construction is discarded, leading to lower retrieval recall and answer accuracy.
> We provide the empirical comparisons below. AssoMem pushes the recall@6 scores **26%, 30%** higher than HippoRAG. Reflecting on the generation scores, we can witness 31% higher than HippoRAG, respectively. Moreover, we highlight the latency and cost here, as in the Table, we can observe that our AssoMem takes around **6x less tokens and 3.4x lower latency** compared to HippoRAG for generation while maintaining a higher accuracy performance.
>
> We analyze these comparisons as such: 1. For HippoRAG, the primary idea is to construct a fine-grained graph via OpenIE through LLM and perform retrieval solely via graph algorithms even if semantic embeddings are used for initialization. This process maximizes the utilization of connectivity information of the graph thus suitable for tackling multi-hop questions with explicit entities appearing. However, there exists information loss in LLM-based OpenIE process and HippoRAG ignores the information from different granularities, on the opposite, answering memory recall questions typically involves semantic relevance or temporal information instead of the connectivity information solely, which leads to the suboptimal results; 2. For A-Mem, it is similar to HippoRAG and further, it summarizes memory and evolves it dynamically, this is ideal for memory updating towards user preference/knowledge etc. And by nature, it’s suitable for answering preference questions instead of recall questions.
>
> | Method    | Recall@3 | Recall@6 | Acc@6 | Avg. Token Consumption for generation | Avg. Latency (s) |
> |-----------|----------|----------|-------|--------------------------------------|------------------|
> | HippoRAG  | 49.66    | 64.18    | 48.61 | 8,430                                | 4.51             |
> | AssoMem   | 72.96    | 80.87    | 64.01 | 1,846                                | 1.30             |
>
> **W2** Several designs (clue generation, temporal token) are bound to chat-based tasks. By comparison, general memory architectures can be applied to and are actually tested on long-context and RAG tasks. [2, 3]
>
> **R2** First of all, we want to discuss that a chat-based memory recall task is also a RAG task, it just focuses on a chat ai scenario, which consists of multi-turn long-term memory. Following this, we provide the result using Long-Context and RAG in the above Table and Table 1 in the paper. We can witness that Long-Context can’t perform well in long-term memory conditions which we attribute this to the fact that large-scale context (all dialogs) for answering memory recall questions typically results in misusing or ignoring of useful memory within long-context, leading to suboptimal performance (multiple existing research verified this as well [1][2]). On the other hand, our AssoMem, featuring a multi-signal scoring system and a fine-tuning strategy, considers information adaptively from different dimensions and makes best use of retrieved content, leading to optimal performance.
>
> | Task Type (LongMemEval_m) | Acc         | Acc (AssoMem) |
> |---------------------------|-------------|---------------|
> | RAG                       | 48.61       | 64.01         |
> | Long-Context              | 21.93       | 64.01         |
>
> **W3** More details regarding the fine-tuning needs to be discussed. It seems that the fine-tuning is an optional component and does not fit with the entire memory architecture. However, it indeed affects the memory read performance a lot, according to Table 2. It is rare to specially design a fine-tuning procedure just to improve the narrow task of “reading chat memory retrieved from a graph index”. Does the fine-tuning also improve the model’s general long-context ability? Conversely, are existing long-context training recipes/models (e.g., [4] [5]) applicable to this paper?
>
> **R3** Yes, the fine-tuning is optional. The motivation behind fine-tuning is the observed gap between retrieval performance and generation performance as detailed in line 387-390 and Appendix B.3. The fine-tuning improves the model’s robustness against retrieval noises. We believe the SFT to improve answer generation against retrieval noises can help general long-context and RAG solutions. We believe [4][5] apply to AssoMem as well but is orthogonal to our FT strategy.

---

> ### Author Response · Authors · 2025-11-22
> **Rebuttal**
>
> **W4** I would like to see more cost analysis both in terms of latency and in terms of tokens consumed.
>
> **R4** Thanks for bringing this to our attention! As shown in the Table and the Table 12, we can explicitly witness that the average latency for a given query to obtain response is 1.3s which is almost **3.4x latency lower** than the faster baseline - MemGAS, which also verified the efficiency of proposed AssoMem. To dive deeper, we further provide the average tokens consumption comparisons, as can be seen, our AssoMem only uses average **1,846** as input to achieve 64.01% accuracy in question answering which further validates that our AssoMem captures the correct memories more efficiently and precisely. The token consumption also contributes to the latency as latency primarily comes from retrieval and generation which again verify the superiority of the complete AssoMem. We sincerely appreciate you bringing this into our attention and will add this part into our paper, expecting to upload it soon as revision.
>
> **W5** I would like to see more details and further analyses to show the value of the new dataset.
>
> **R5** Thanks for this review. Dataset details: the MeetingQA dataset, till the paper submission date, is a collection of 50 voice memos, each containing approximately 70–80 messages, with a variety of speaker configurations (ranging from 2 to 6 speakers per memo). It includes 390 Q&A pairs, where each answer is mapped to at least one specific message within a corresponding voice memo.
> Analysis:
>
> the proposed MeetingQA has the following values.
>
> It is the first dataset mimicking an important scenario–long meetings w. multi-turn dialogs from different speakers abundant.
> The dataset serves as a good testbed for scenarios where retrieval correctness is crucial for achieving high accuracy in answering memory recall questions.
>
> Long-context/RAG/Long-term memory tasks can also benefit from this dataset as it includes multi-turn, multi-speaker corpus which is close to real-world scenarios but harder for existing methods to tackle.
>
> We are consistently developing this dataset even after the paper submission, and now we have more high-quality data out as below:
>
> The cumulative new dataset comprises 314 distinct voice memo sessions and approximately 1,790 Question-Answer pairs. The data features high-density conversation structures (averaging ~70–80 turns per session) with speaker configurations ranging from 2 to 6 participants. The composition is stratified as follows:
>
> Real-World Data: **89 authentic voice memos paired with ~500 expert-annotated Q&A pairs.**
>
> Domain-Specific Data: **75 "David AI" sessions with ~500 expert-annotated Q&A pairs.**
>
> Synthetic Data: **150 LLM-generated sessions (50 baseline + 100 enhanced) paired with ~790 Q&A pairs.**
> memory recall. We’ve already shared the dataset in the paper as in this anonymous link: https://anonymous.4open.science/r/AssoMem-BDB7,
>
> **Reference**
>
> [1] Lost in the Middle: How Language Models Use Long Contexts
>
> [2] CRAG-MM: Multi-modal Multi-turn Comprehensive RAG Benchmark

---

> ### Comment · Reviewer_xRKF · 2025-11-25
>
> Thank you for your response and additional experiments. I have raised my score. Please incorporate the new results into the paper to improve its completeness.
>
> On weakness 2, I actually was thinking of typical non-chat tasks that [2, 3] tested on, such as multi-hop retrieval-augmented QA (e.g., Musique) and long-context QA (e.g., Qasper). Chat memory is indeed RAG and long-context but my impression is that it is only a small part of the picture.

---

> > ### Author Response · Authors · 2025-11-25
> >
> > Thank you! We agree that Chat memory is part of long-context RAG tasks. We appreciate your insights on this, it would be interesting to test on the datasets you provided, we'll keep you updated!

---

### Official Review · Reviewer_P9dL · 2025-10-20

**Soundness:** 2
**Presentation:** 2
**Contribution:** 2
**Rating:** 4
**Confidence:** 3

**Summary:**

This paper introduces AssoMem, a memory-augmented question answering framework that constructs an associative memory graph to link dialogue utterances with automatically extracted clues. By combining multi-dimensional retrieval signals—relevance, importance, and temporal alignment—using a mutual information-driven fusion strategy, AssoMem achieves importance-aware and context-sensitive memory recall. Extensive experiments across three benchmarks, including the newly proposed MeetingQA dataset, demonstrate its superiority.

**Strengths:**

1. AssoMem integrates relevance, importance, and temporal alignment effectively, enabling context-rich and adaptive memory recall for QA tasks.

2. The use of an associative memory graph is a contribution, facilitating semantic connections and improving memory organization.

3. Introduction of the MeetingQA benchmark and extensive experiments validate AssoMem's superiority, with clear performance gains over prior methods.

**Weaknesses:**

1. The innovation of the paper appears to be limited, as the idea of capturing relationships between memories using graph representations has already been explored by several existing methods, such as HippoRAG [1] and A-Mem [2].

2. The experimental comparisons in the paper seem insufficient. In addition to the aforementioned methods, the authors have not compared their approach with other well-known memory-based methods, such as mem0 [3].

3. I find the model fine-tuning process described in Section 3.3 confusing. For QA tasks, wouldn't prior fine-tuning risk leaking information about subsequent queries? Furthermore, such a fine-tuning process seems to lack practical application value in real-world scenarios.

4.  The construction details of the MeetingQA dataset are insufficiently described, and the dataset has not been made publicly available. This raises concerns about the fairness and reliability of the dataset.

5. Some formulas in the paper lack clear explanations of their symbols. For instance, the meaning of $y_m^\lambda$ in Eq. (3) is not adequately clarified.

[1] Jimenez Gutierrez, Bernal, et al. "Hipporag: Neurobiologically inspired long-term memory for large language models." Advances in Neural Information Processing Systems 37 (2024): 59532-59569.

[2] Xu, Wujiang, et al. "A-mem: Agentic memory for llm agents." arXiv preprint arXiv:2502.12110 (2025).

[3] Chhikara, Prateek, et al. "Mem0: Building production-ready ai agents with scalable long-term memory." arXiv preprint arXiv:2504.19413 (2025).

**Questions:**

See weakness.

---

> ### Author Response · Authors · 2025-11-22
> **Rebuttal**
>
> **In Responding to Weakness**
>
> **W1** The innovation of the paper appears to be limited, as the idea of capturing relationships between memories using graph representations has already been explored by several existing methods, such as HippoRAG [1] and A-Mem [2].
>
> **R1** Comparisons:
>
> | Method    | Recall@3 | Recall@6 | Acc@6 | Avg. Token Consumption | Avg. Latency (s) |
> |-----------|----------|----------|-------|-----------------------|------------------|
> | HippoRAG  | 49.66    | 64.18    | 48.61 | 8,430                 | 4.51             |
> | A-Mem     | 47.82    | 62.17    | 48.06 | 8,852                 | 2.45             |
> | AssoMem   | 72.96    | 80.87    | 64.01 | 1,846                 | 1.30             |
>
> AssoMem achieves 46.9% and **52.6% improvements in Recall@3 compared to HippoRAG and A-Mem on LongMemEval_m, respectively. We attribute this to the fact that AssoMem integrates multi-dimensional signals with adaptively weight assignments for better memory record retrieval, enhancing its recall capabilities. AssoMem also outperforms HippoRAG and A-Mem in Recall@6 by **26.0% and 30.0%**, respectively, and in Acc@6 by **31.7% and 33.2%**, respectively. These improvements underscore AssoMem's superior performance and efficiency.
>
> P.S. We encounter multiple errors during implementing Mem0, will keep you posted once we have results.
>
> **W3** I find the model fine-tuning process described in Section 3.3 confusing. For QA tasks, wouldn't prior fine-tuning risk leaking information about subsequent queries? Furthermore, such a fine-tuning process seems to lack practical application value in real-world scenarios.
>
> **R3** Thanks for this review. The SFT would not lead to information leaking for two reasons. First, as detailed in Section 3.3, the fine-tuning process explicitly separates the queries into training set and testing set, so there would be no risk leaking information about subsequent queries. Second, in the dataset, each qa-pair is associated with a different set of dialogue sessions.
>
> The intuition of fine-tuning strategy is that we observed that the recall performance improvement does not always transfer to generation performance due  to the introduction of noises (rare existing memory works consider this) ( line 376-396 and Appendix B.3).  Our experiments show that SFT can improve the QA results by **8%** higher on average.
>
> **W4** The construction details of the MeetingQA dataset are insufficiently described, and the dataset has not been made publicly available. This raises concerns about the fairness and reliability of the dataset.
>
> **R4** Thanks for the review. We detailed data construction in Appendix A: We first endorse different personas into  an LLM (2-6 speakers per meeting) and then seed a subset of real-world topics (50 topics) for them to chat, mimicking the meeting process. To assure the quality, we employ a linguistic engineer to manually check the generation quality and curate question-answer pairs for each meeting. We provide a subset of samples in this anonymous repo here: https://anonymous.4open.science/r/AssoMem-BDB7. We’re consistently investigating this dataset, now containing real meeting records and new generated data with two linguistic engineers to manually check the quality. We’re planning to release it later to the research community. You may also refer to **R3** to Reviewer xUZR.
>
> **W5** Some formulas in the paper lack clear explanations of their symbols. For instance, the meaning of in Eq. (3) is not adequately clarified.
>
> **R5** Sorry for the confusion, as detailed in line 236-245:
>
> \hat(s)_{u}^{d}{b} denotes the score of memory u from dimension b;
>
> \lambda is the usefulness label,
>
> q is the query
>
> Eq (3) is for computing the conditional mutual information, the high-level idea is to compute the conditional probability of score from each dimension in terms of given query q and further, compute the conditional mutual information based on each conditional probability on each dimension. The CMI function can be further used to adaptively assign weights when a new query comes. Please do not hesitate to point out anything in Eq (3) that confuses you, we’d be happy to address them for you here!

---

### Official Review · Reviewer_xUZR · 2025-10-29

**Soundness:** 3
**Presentation:** 2
**Contribution:** 2
**Rating:** 4
**Confidence:** 4

**Summary:**

The paper proposes AssoMem, a memory question-answering framework designed to improve large-scale memory retrieval for LLM-based assistants. The key idea is to model associative human-like recall through a multi-signal retrieval mechanism that integrates relevance, importance, and temporal dimensions. It constructs an associative memory graph linking utterances to automatically extracted ”clues“，employs Personalized PageRank for importance scoring, and fuses multi-dimensional scores using a mutual information–based weighting strategy, with a specific fine-tuning strategy for the LLM. In addition, the paper also introduce new benchmark, MeetingQA, to simulate real-world meeting scenarios where multi-turn dialogues form the memory base paired with diverse QA examples. Experiments on LongMemEval and a new synthetic benchmark, MeetingQA, show consistent improvements over baselines.

**Strengths:**

1. The benchmark is valueable.
2. The proposed method is effective as valied by comprehensive experiments.

**Weaknesses:**

1. The method is overly complex yet underexplained. There are many components in the proposed method, i.e., associative memory graph,  multi-dimensional scores fusion and specific fine-tuning strategy. Desipte putting them together leads to better performance, it is unclear current design is the best way or it is worthy. For example, what if combine MemGAS with later multi-dimensional scores fusion? What if do not finetune the model?
2. The associative memory graph construction and page rank part is quite similiar with MemGAS, which weaken the contribution in the method side.
3. Some details or experiments are not clear. for example, how to make sure the quality and diversity of introduced benchmark? the detailed latency comparison with other baselines.
4. The paper can benefit with improved writing and presentation. The text is technically overloaded, using jargon and formulaic notations without sufficient narrative clarity. Figures and examples provide limited intuition.

**Questions:**

See weakness

---

> ### Author Response · Authors · 2025-11-22
> **Rebuttal**
>
> Thanks for reviewing our work!
>
> **In Responding to Weakness**
>
> **W1** The method is overly complex yet underexplained. There are many components in the proposed method, i.e., associative memory graph, multi-dimensional scores fusion and specific fine-tuning strategy. Despite putting them together leads to better performance, it is unclear current design is the best way or it is worthy. For example, what if combine MemGAS with later multi-dimensional scores fusion? What if do not finetune the model?
>
> **R1** We have presented the results of non-fine tuning vs fine-tuning the models from a range of size options as in line 376-396 and Table 2. For fine-tuning, as observed in Table 2, on LlaMA3.2-3B-Instruct, non-fine tuning achieves an accuracy result of 26.91% while after our fine-tuning, the results is 33.43%; on Qwen2.5-32B-Instruct, non-fine tuning achieves an accuracy result of 64.72% while after our fine-tuning, the results is 73.88%. An average **8%** improvements are brought by fine-tuning.
>
> Adjunct with Ablation Study in Section 5.2, without our proposed CMI strategy, the accuracy reduces from 69.17% (full AssoMem) to 58.89%; without temporal signal, the accuracy reduces from 69.17% to 61.19%; without importance signal, the accuracy reduces from 69.17% to 61.97%. These observations along with the fine-tuning results verify the necessity of each component of AssoMem.
>
> For anything else considered underexplained, We're committed to addressing it by revisions.
>
> **W2** The associative memory graph construction and page rank part is quite similar with MemGAS, which weaken the contribution in the method side.
>
> **R2** Although we have similar graph construction process, we highlight a few major differences:
>
> First, the purpose of using pagerank is different: MemGAS mainly runs PPR to adaptively assign weights on **different granularities** while our AssoMem uses the PPR score as importance scores, involving in the final retrieval step;
>
> Secondly, MemGAS perform memory retrieval solely using semantic similarity while our AssoMem integrates signals from **relevance, importance and temporal**;
>
> Thirdly, we step further using conditional mutual information which considers the likelihood of the usefulness of memory candidates w.r.t given query while MemGAS consider solely on the memory base information entropy.
>
> As a result, our AssoMem makes improvements of **14%, 10%** on retrieval recall, and generation accuracy against MemGAS, respectively.
>
> **W3** Some details or experiments are not clear. for example, how to make sure the quality and diversity of introduced benchmark? the detailed latency comparison with other baselines.
>
> **R3** Thanks for the review. As detailed in Appendix A, the MeetingQA dataset  is a collection of 50 meetings, each containing approximately messages, with a variety of speaker configurations (ranging from 2 to 6 speakers per memo). We have linguistic engineers with a background in linguistics and LLM to perform human evaluations to ensure the quality. Even after this submission, the dataset is continuing to expand and more evaluations are done for the quality, the expanded MeetingQA dataset contains:
>
> Real-World Data: **89 authentic voice memos paired with ~500 expert-annotated Q&A pairs.**
>
> Domain-Specific Data: **75 "David AI" sessions with ~500 expert-annotated Q&A pairs.**
>
> Synthetic Data: **150 LLM-generated sessions (50 baseline + 100 enhanced) paired with ~790 Q&A pairs.**
>
> We also provide a subset of samples in this anonymous repo here: https://anonymous.4open.science/r/AssoMem-BDB7. We will release the dataset for the research community.
>
> **Latency Comparisons**
>
> | Method    | Avg. Tokens Consumed For Generation | Avg. Latency (s) | Acc    |
> |-----------|-------------------------------------|------------------|--------|
> | HippoRAG  | 9,098                               | 4.51             | 48.06  |
> | MemGAS    | 1,846                               | 2.45             | 64.01  |
> | A-Mem     | 2,073                               | 2.59             | 62.50  |
> | AssoMem   | 2,073                               | 1.30             | 62.50  |
>
> As shown in the Table and the Table 12, we can explicitly witness that the average latency for a given query to obtain response is 1.3s which is almost **88%** faster than the faster baseline - MemGAS, verifying the efficiency of proposed AssoMem. To dive deeper, we further provide the average tokens consumption comparisons, as can be seen, our AssoMem only uses average **1,846** as input to achieve 64.01% accuracy in question answering, which further validates that our AssoMem captures the correct memories more efficiently and precisely. The token consumption also contributes to the latency as latency primarily comes from retrieval and generation which again verify the superiority of the complete AssoMem.
>
> We sincerely appreciate you bringing this into our attention and will add this part into our revision, expecting to upload it soon.

---

> > ### Comment · Reviewer_xUZR · 2025-11-25
> >
> > Thank you for your clarification. I decide to raise my initial score, please make sure incorporating these explanation into your next version and refine the presentation.

---

> > > ### Author Response · Authors · 2025-11-25
> > >
> > > Thank you for your reviews and suggestions! We'll include our discussions into revision and upload it soon!

---

### Official Review · Reviewer_1BTS · 2025-11-01

**Soundness:** 2
**Presentation:** 2
**Contribution:** 2
**Rating:** 4
**Confidence:** 4

**Summary:**

This paper proposes a novel memory QA framework, AssoMem, designed to address the challenge of accurate recall for AI assistants operating on large-scale, similarity-dense memory repositories. Existing methods, which rely primarily on semantic relevance between the query and memory, perform poorly in these "similarity-dense" scenarios. Inspired by human associative memory, AssoMem constructs an "associative memory graph" that anchors dialogue utterances to automatically extracted "clues." This graph structure enables "importance-aware" ranking of memories. The core of the framework is a retriever called RITRanker, which innovatively integrates three distinct signals: Relevance, Importance, and Temporal alignment. These signals are weighted using an adaptive fusion strategy driven by CMI to dynamically adjust each signal's contribution based on the query's intent. The paper also introduces a new benchmark, MeetingQA. Experiments demonstrate that AssoMem outperforms existing baselines across several datasets.

**Strengths:**

1. The paper correctly identifies a core weakness in existing RAG and memory systems: relevance-only retrieval is ineffective in "similarity-dense" scenarios.
2. The introduction of the MeetingQA dataset is a useful contribution to the community, addressing the need for benchmarks that simulate real-world, multi-turn dialogue scenarios.

**Weaknesses:**

1. The paper claims scalability, but the offline graph construction (which includes LLM-based tagging for every session) takes ~1950 seconds (Table 12). This is for a small dataset of 2,500 sessions. When the memory scales to the 100k or 1M entries expected of a "second brain," this offline cost seems prohibitively expensive, challenging the "scalable" claim.
2. The "importance" score relies on running Personalized PageRank at query time (Section 3.2.3, "the utterance cells in t are set to the similarity between query and utterance"). This requires a graph computation for every query, which can be slow on large graphs. The paper does not analyze this specific query latency, only the overall "RIT Scoring" (0.39s).
3. The paper bundles AssoMem with a generator LLM fine-tuning strategy (Section 3.3). This conflates the contributions. The main results in Table 1 represent the full system (including the fine-tuned LLM). This comparison is potentially unfair, as the baseline methods (e.g., "Topic grouping") do not appear to receive this specialized denoising and multi-task fine-tuning. The performance gain attributed to AssoMem might stem not just from the superior retriever but also from a specially-tuned generator.
4. A key innovation is the adaptive CMI fusion. However, the ablation study (Table 10) comparing it to other fusion strategies (LR, RF, SVM) is relegated to the appendix. More importantly, it is missing the most crucial baseline: a simple, non-adaptive weighted sum.

**Questions:**

1. The CMI fusion strategy appears to require (query, utterance, usefulness label) triples for training. How is this model trained? Does this imply AssoMem must be deployed only after a significant data annotation effort for a specific user, or is it trained once on the benchmark datasets and then generalized?
2. The paper mentions using query-personalized PPR. What is the actual latency of this PPR computation on the large graph (LongMemEval_l), and how does this latency scale as the graph grows from 2.5k to 100k sessions?

*Typo:
Line 82: Associative memory graph -> Associative memory graph:

---

> ### Author Response · Authors · 2025-11-22
> **Rebuttal**
>
> We sincerely thank you for taking the time to review our work and recognize the value of the insightful comments you have made. We are committed to addressing the issues you have noted and endeavoring to assure the quality of this study.
>
> **In Responding to your weakness**
>
> **W1** The paper claims scalability, but the offline graph construction (which includes LLM-based tagging for every session) takes ~1950 seconds (Table 12). This is for a small dataset of 2,500 sessions. When the memory scales to the 100k or 1M entries expected of a "second brain," this offline cost seems prohibitively expensive, challenging the "scalable" claim.
>
> **R1** Thanks for this review. We understand your concern here and would like to make some clarifications:
>
> 1. **We are faster than methods in related work.** As detailed in Table 5 (Dataset statistics), 2,500 sessions yield 2,500*9.76≈24.4k utterances which means for a graph we have 24.4k nodes. The offline construction takes around 1,950 seconds which is around **23% less** than the construction latency of 2,398 seconds of MeGAS[2] (They didn’t report this number in the original paper, we calculate it based on the open-reproduction here: https://github.com/quqxui/MemGAS/blob/main/src/construct/construct_asso.py).
>
> 2. 2500 dialogue sessions are NOT small. For example, a user may typically have 2.8 [1] sessions with ChatGPT which lead to around 893 days (2.5 years) for reaching 2,500 sessions.
>
> 3. Even when there are significantly higher number of sessions in future, graph construction will happen **incrementally** not built at once. Incrementally adding a node takes only **0.01** second as provided in Table 12.
>
> Hope these explanations help you better understand the latency and reduce your concerns here!
>
> **W2** The "importance" score relies on running Personalized PageRank at query time (Section 3.2.3, "the utterance cells in t are set to the similarity between query and utterance"). This requires a graph computation for every query, which can be slow on large graphs. The paper does not analyze this specific query latency, only the overall "RIT Scoring" (0.39s).
>
> **R2** Thanks for the concern here! We want to elaborate that the “RIT Scoring" includes the similarity comparisons, personalized pagerank computation (PPR) and temporal similarity comparisons. the latency of PPR solely is 0.21 (lower than 0.39) .
>
> Even though  the graph may grow over time, we can apply a recency threshold to run PPR on only recent nodes, to assure the latency while maintaining high performance.
>
> **W3** The paper bundles AssoMem with a generator LLM fine-tuning strategy (Section 3.3). This conflates the contributions. The main results in Table 1 represent the full system (including the fine-tuned LLM). This comparison is potentially unfair, as the baseline methods (e.g., "Topic grouping") do not appear to receive this specialized denoising and multi-task fine-tuning. The performance gain attributed to AssoMem might stem not just from the superior retriever but also from a specially-tuned generator.
>
> **R3** Sorry for the confusion. We would like to clarify that all methods sreported in Table 1 use **Llama3.3-70B-Instruct for fair comparisons.** AssoMem is not fine-tuned, as indicated in Section 4.3, to present the effectiveness of AssoMem and RIT Scoring system
>
> We investigated the effectiveness of fine-tuning in Table 2, where we witnessed **8%** improvements by the fine-tuning strategy. A key innovation is the adaptive CMI fusion. However, the ablation study (Table 10) comparing it to other fusion strategies (LR, RF, SVM) is relegated to the appendix. More importantly, it is missing the most crucial baseline: a simple, non-adaptive weighted sum.
> we understand your concerns and we attach results comparing to non-adaptive weight assignment
>
> | Fusion strategy                                   | Recall@6 | Recall@10 | Acc@6  |
> |---------------------------------------------------|----------|-----------|--------|
> | Non-adaptive weight assignment (0.6*R+0.3*I+0.1*T)| 76.7     | 80.80     | 59.38  |
> | SVM                                               | 78.81    | 81.66     | 62.11  |
> | Information gain                                  | 80.94    | 86.33     | 63.94  |
> | CMI                                               | 82.64    | 88.91     | 64.20  |
>
> As in the table, “0.6*R+0.3*I+0.1*T” denotes the non-adaptive weighted sum for RIT scoring - 0.6*relevance plus 0.3*importance plus 0.1*temporal to obtain the final score for retrieval across the dataset. The combination selection is the best result from experimental findings. CMI improved over this fixed weight assignment by **9%**.

---

> ### Author Response · Authors · 2025-11-22
> **Rebuttal**
>
> **In Responding to Questions**
>
> **Q1** The CMI fusion strategy appears to require (query, utterance, usefulness label) triples for training. How is this model trained? Does this imply AssoMem must be deployed only after a significant data annotation effort for a specific user, or is it trained once on the benchmark datasets and then generalized?
>
> **A1** As detailed in line 230-259, CMI is a statistical method that does not require significant data annotation efforts: it  measures between two variables, X and Y, how much information X and Y share, given that we already know Z.
>
> Indeed, we highlight that this is actually a **benefit** of using CMI because: 1. The usefulness labels are super unbalanced, imagine for each user’s query, among the 24.4k utterances, only [1, 4] of them are memory evidence for answering this query. The unbalanced labels pose challenges to do the training even with upper sampling and down sampling as detailed in Table 10 and Appendix B.5; 2. CMI directly measures how well a signal from each dimension reflects the likelihood that a memory utterance is useful for answering a given query which theoretically is more suitable for this task and experimentally verified in Table 10.
>
> **Q2** The paper mentions using query-personalized PPR. What is the actual latency of this PPR computation on the large graph (LongMemEval_l), and how does this latency scale as the graph grows from 2.5k to 100k sessions?
>
> **A2** The actual latency of PPR in runtime is ~0.21s, and this is the latency on LongMemEval_l. 100k sessions are not typically applicable in existing benchmarks and we conduct a data augmentation to sample more dialog sessions to 100k for experiments and the latency is: 51.9s, running on a cpu node configured with 16 cores and 200G memory. We also want to highlight that as indicated in the response to weakness 2, a strategy here for such a large graph is to run PPR on a recent sub-graph to assure latency while maintaining performance.
>
> *Typo: Line 82: Associative memory graph -> Associative memory graph:
>
> Thanks for pointing this out, we will enhance our writings in the revision.
>
> **Reference**
>
> [1]https://www.zebracat.ai/post/chatgpt-usage-statistics#:~:text=Enterprise%20users%20average%204.5%20sessions%20per%20workday%2C,ChatGPT%20for%20personal%20learning%20or%20skill%20development.
>
> [2] FROM SINGLE TO MULTI-GRANULARITY: TOWARD LONG-TERM MEMORY ASSOCIATION AND SELECTION OF CONVERSATIONAL AGENTS

---

### Author Response · Authors · 2025-12-01
**Statement regarding recent incident and summary of rebuttal updates**

Dear Reviewers, ACs, SACs, and PCs,

We sincerely appreciate the time and effort you have dedicated to reviewing our work. In light of the recent information leakage incident, we are committed to full transparency and wish to provide a formal statement regarding our conduct, followed by a brief summary of our rebuttal updates to facilitate a better understanding of our work.

**Statement of Integrity**

**No Contact:** We confirm that following the incident, we made no attempts to search for or reach out to our reviewers. All interactions on our end have remained strictly within this page and have never violated the double-blind policy.

**Timeline of Ratings:** Please note that the **rating increases (4->6)** from Reviewer xUZR and Reviewer xRKF occurred **before** the incident, as verified by the system history logs. We believe our efforts and explanations addressed their concerns, leading to the increased ratings.

To assist in the further review process for the new AC, we would like to make some **highlights** as below:

**Key Innovations compared with SOTA**
* Our AssoMem featured with a **topic-clue structured graph** and an **adaptive weights assignments** which enable the considerations of underlying associative patterns between question types and information from **different granularities**.
* The **key difference** can be concluded as two-fold:
  * Graph construction: Unlike existing graph-based memory methods (e.g., MemGAS, HippoRAG) that utilize fine-grained and abstractive entity nodes (e.g., keywords, summaries, whole sessions), our graph is more lightweight.
  * RIT scoring system: Unlike most memory methods using relevance solely for retrieval, our AssoMem adaptively uses signals from semantic relevance, PPR popularity and temporal dimensions for final scoring and retrieval.
* The benefits of such structure can be:
  * **Performance** - AssoMem outperforms existing SOTA memory methods (e.g., MemGAS, A-Mem) with an average improvement of 14% on retrieval and 10\% on qa performance.

  * **Scalability** - Attributing to the lightweight graph and advanced scoring system, AssoMem achieves approximately **3x lower latency** and **8x lower token cost** compared to SOTA methods while maintaining competitive performance.

* We have added the comparisons and new comparative study results into the revision.


We also summarized the primary concerns raised and how we addressed them to achieve the improved ratings:

**Most Common Concerns & our Rebuttal**:

**Concern 1:** Quality and details of the proposed MeetingQA dataset.

**Response:** We have provided detailed descriptions and statistics in Appendix A. Furthermore, we introduced new human-annotated and synthetic data—verified by linguistics engineers—to substantiate the contributions of MeetingQA.

**Concern 2:** Latency and cost of the proposed AssoMem.

**Response:** We conducted detailed comparisons with existing well-known methods. The results demonstrate that AssoMem achieves approximately **3x lower latency** and **8x lower token cost** compared to SOTA methods while maintaining competitive performance. We will include these results in our revision.

**Concern 3:** Clarification on the fine-tuning strategy.

**Response:** Reviewers raised concerns regarding the purpose and generalizability of our fine-tuning strategy. We clarified that this strategy is designed to reduce noise introduced by retrieved memories. It does not create unfair comparisons, as it is evaluated separately (see Table 2). Furthermore, our results show that this strategy also benefits Long-context and RAG tasks.

We sincerely appreciate the constructive feedback and acknowledge that we have addressed additional points (such as baseline comparisons) in the full discussion threads as below (feel free to check more). We respectfully ask the ACs, SACs, and PCs to consider our detailed responses and the resulting rating increases in their final meta-reviews and decisions.

---

### Meta-Review · Area_Chair_t5Cd · 2026-01-06

**Summary:**

The paper presents AssoMem, a framework designed to enhance memory-augmented QA by constructing an associative memory graph and utilizing a multi-signal retrieval system (RITRanker). The framework addresses similarity-dense scenarios where standard RAG systems struggle. Initially, reviewers were uncertain due to concerns regarding the scalability: The computational cost of graph construction and PageRank at scale; Novelty: Overlap with existing graph-based methods like HippoRAG and MemGAS; Methodological Clarity: The role of the Supervised Fine-Tuning (SFT) strategy and whether it created an unfair comparison, and Benchmark Quality: The reliability and availability of the new MeetingQA dataset.

During the rebuttal, the authors provided extensive quantitative evidence and clarifications that effectively shifted the consensus toward acceptance.

**Reviewer Concerns:**

Addressed concerns:
- Authors clarified that the primary results (Table 1) used a frozen Llama-3-70B model for all methods, ensuring the gains were attributed to the AssoMem retriever and not the specialized fine-tuning
- The authors provided a detailed cost-latency analysis. AssoMem is significantly faster (1.3s vs. HippoRAG's 4.51s) and more token-efficient (~1.8k vs. ~8.4k tokens) than state-of-the-art baselines
- Authors demonstrated that while full graph construction takes time, incremental updates (adding new sessions) take only 0.01s, making it viable for long-term use
- Authors argued that AssoMem's use of utterance-level nodes (vs. HippoRAG’s entity nodes) avoids Information Extraction (IE) loss, while the CMI-driven fusion outperforms standard weighted sums by 9%

Outstanding concerns:
- Reviewer xRKF noted that while AssoMem excels at chat-based memory recall, its applicability to broader, non-conversational RAG tasks (e.g., multi-hop reasoning over documents) remains less explored compared to general architectures
- Although the recency sub-graph strategy was proposed, the raw latency of Personalized PageRank on extremely large graphs (100k+ sessions) without such pruning remains a potential bottleneck for edge-case deployments

**Reviewer Scores:**

For reviewer xUZR and reviewer xRKF, they explicitly raised score from negative to positive after their concerns are solved. For the other two reviewers, their concerns about CMI vs. fixed weights, scalability, and the HippoRAG/A-Mem comparative results are partially soveld, and their score might unchange or move to marginal positive.

---

### Decision · Program_Chairs · 2026-01-26

Accept (Poster)